# A B-cell actomyosin arc network couples integrin co-stimulation to mechanical force-dependent immune synapse formation

Jia C Wang[1], Yang-In Yim[1], Xufeng Wu[2], Valentin Jaumouille[1], Andrew Cameron[1], Clare M Waterman[1], John H Kehrl[3], John A Hammer[1]*

[1]Cell and Developmental Biology Center, National Heart, Lung and Blood Institute, National Institutes of Health, Bethesda, United States; [2]Light Microscopy Core, National Heart, Lung and Blood Institute, National Institutes of Health, Bethesda, United States; [3]B Cell Molecular Immunology Section, National Institutes of Allergy and Infectious Diseases, National Institutes of Health, Bethesda, United States

*For correspondence:
hammerj@nhlbi.nih.gov

Competing interest: The authors declare that no competing interests exist.

**Abstract** B-cell activation and immune synapse (IS) formation with membrane-bound antigens are actin-dependent processes that scale positively with the strength of antigen-induced signals. Importantly, ligating the B-cell integrin, LFA-1, with ICAM-1 promotes IS formation when antigen is limiting. Whether the actin cytoskeleton plays a specific role in integrin-dependent IS formation is unknown. Here, we show using super-resolution imaging of mouse primary B cells that LFA-1:ICAM-1 interactions promote the formation of an actomyosin network that dominates the B-cell IS. This network is created by the formin mDia1, organized into concentric, contractile arcs by myosin 2A, and flows inward at the same rate as B-cell receptor (BCR):antigen clusters. Consistently, individual BCR microclusters are swept inward by individual actomyosin arcs. Under conditions where integrin is required for synapse formation, inhibiting myosin impairs synapse formation, as evidenced by reduced antigen centralization, diminished BCR signaling, and defective signaling protein distribution at the synapse. Together, these results argue that a contractile actomyosin arc network plays a key role in the mechanism by which LFA-1 co-stimulation promotes B-cell activation and IS formation.

## Editor's evaluation

This study has used striking live-cell super-resolution microscopy methods to demonstrate the direct relationship between integrin dependent F-actin/myosin arcs and transport of surface immunoglobulin-antigen clusters in B-cell synapses. The function of the F-actin arcs in signaling at limiting antigen levels where integrins are important is also demonstrated. In addition, this study has shown that both follicular and germinal center B cells utilize the F-actin arcs, suggesting that this machinery can operate in both initiation and affinity maturation phases of antibody-based adaptive immunity. The work will be of interests to immunologists, cell biologists, and biophysicists, and the data sets should be of future use in modeling the process.

## Introduction

B-cell receptor (BCR) engagement with cognate antigen triggers striking changes in B-cell physiology that promote B-cell activation, immune synapse (IS) formation, and B-cell effector functions (**Forthal,**

**eLife digest** The immune system has the ability to recognize a vast array of infections and trigger rapid responses. This defense mechanism is mediated in part by B cells which make antibodies that can neutralize or destroy specific disease-causing agents. When pathogens (such as bacteria or viruses) invade the body, a specialized immune cell called an 'antigen presenting cell' holds it in place and presents it to the B cell to examine. Receptors on the surface of the B cell then bind to the infectious agent and launch the B cell into action, triggering the antibody response needed to remove the pathogen.

This process relies on B cells and antigen presenting cells making a close connection called an immune synapse, which has a bulls-eye pattern with the receptor in the middle surrounded by sticky proteins called adhesion molecules. A network of actin filaments coating the inside of the B cell are responsible for arranging the proteins into this bulls-eye shape. Once fully formed, the synapse initiates the production of antibodies and helps B cells to make stronger versions of these defensive proteins.

So far, most studies have focused on the role the receptor plays in B cell activation. However, when there are only small amounts of the pathogen available, these receptors bind to the antigen presenting cell very weakly. When this happens, adhesion molecules have been shown to step in and promote the formation of the mature synapse needed for B cell activation. But it is not fully understood how adhesion molecules do this.

To investigate, Wang et al. looked at mouse B cells using super resolution microscopes. This revealed that when B cells receive signals through both their receptors and their adhesion molecules, they rearrange their actin into a circular structure composed of arc shapes. Motors on the actin arcs then contract the structure inwards, pushing the B cell receptors into the classic bullseye pattern. This only happened when adhesion molecules were present and signals through the B cell receptors were weak.

These findings suggest that adhesion molecules help form immune synapses and activate B cells by modifying the actin network so it can drive the re-patterning of receptor proteins. B cells are responsible for the long-term immunity provided by vaccines. Thus, it is possible that the findings of Wang et al. could be harnessed to create vaccines that trigger a stronger antibody response.

---

*2014*; *Heesters et al., 2016*; *Harwood and Batista, 2011*). These changes include dramatic increases in actin filament assembly and dynamics that are thought to drive IS formation in B cells engaged with membrane-bound antigen (*Forthal, 2014*; *Heesters et al., 2016*; *Harwood and Batista, 2011*; *Gonzalez et al., 2011*). For B cells in vivo, this usually involves interactions with antigen bound to the surface of an antigen-presenting cell (APC) (*Gonzalez et al., 2011*; *Carrasco and Batista, 2006a*; *Cyster, 2010*), although activating surfaces such as antigen-coated glass and planar lipid bilayers (PLBs) containing freely diffusing antigen are used to mimic these in vivo interactions. IS formation in these contexts is initiated by the formation of a radially symmetric, Arp2/3 complex-dependent branched actin network at the outer edge of the IS (i.e., in the distal supramolecular activation cluster [dSMAC]) (*Wang and Hammer, 2020*; *Song et al., 2014*). This lamellipodia-like actin network drives the spreading of the B cell across the antigen-coated surface, thereby promoting BCR:antigen interactions (*Harwood and Batista, 2011*; *Fleire et al., 2006*). Once the B cell is fully spread, the continued polymerization of branched actin at the outer edge of the dSMAC generates a centripetal or retrograde flow of actin that drives the movement of BCR:antigen clusters (*Tolar et al., 2009*; *Mattila et al., 2016*; *Treanor et al., 2009*) towards the center of the synapse (i.e., to the central SMAC [cSMAC]) (*Song et al., 2014*; *Bolger-Munro et al., 2019*). This centripetal actin flow, combined with an overall contraction of the B cell, is thought to be responsible for the transport of BCR:antigen clusters to the center of the maturing synapse (*Bolger-Munro et al., 2019*). Importantly, this process of antigen centralization is required for robust BCR signaling (*Harwood and Batista, 2011*; *Mattila et al., 2016*; *Batista et al., 2010*) and is thought to be a prerequisite for antigen internalization by follicular B cells (*Batista et al., 2001*; *Yuseff et al., 2013*; *Yuseff and Lennon-Duménil, 2015*).

Antigen-induced IS formation scales with the strength of antigen-induced signals such that IS formation and B-cell activation are attenuated when membrane-bound antigen binds the BCR weakly or is

presented at low density. Importantly, co-stimulatory signals can promote IS formation and B-cell activation under both of these conditions (*Carrasco et al., 2004*; *Carrasco and Batista, 2006b*). Seminal work from Carrasco and colleagues showed that the B-cell integrin LFA-1, which binds the adhesion molecule ICAM-1 present on the surface of APCs (*Springer, 1990*; *Springer et al., 1987*), serves as one such co-stimulatory signal (*Carrasco et al., 2004*). This conclusion was based on four key observations. First, B cells responded robustly to higher affinity membrane-bound antigens presented at high density whether or not ICAM-1 was present on the membrane. Second, the robust activation of B cells in response to antigens of all affinities increasingly required ICAM-1 in the membrane as the density of the antigen was lowered. Third, this co-stimulatory effect was most dramatic for weaker antigens. Finally, this latter effect was not observed in B cells lacking LFA-1. With regard to the underlying mechanism, IRM imaging suggested that LFA-1:ICAM-1 interactions, which were shown to concentrate in the medial portion of the synapse (i.e., the peripheral SMAC [pSMAC]), lower the threshold for B-cell activation by enhancing cell adhesion.

While the actin cytoskeleton clearly plays a central role in driving IS formation, whether it plays a specific role in integrin-dependent IS formation is unknown. This is an important question as most B-cell interactions with professional APCs presenting cognate antigen involve integrin ligation. Relevant to this question, the dendritic actin network occupying the outer dSMAC ring, which is thought to be the main driver of IS formation, has been observed primarily in cells that received antigen stimulation alone and almost exclusively in immortalized B cell lines (*Bolger-Munro et al., 2019*; *Wang et al., 2018*; *Freeman et al., 2011*; *Liu et al., 2012*; *Wang et al., 2017*). It is not known, therefore, whether integrin-co-stimulation alters the organization and/or dynamics of actin at the B-cell IS. Moreover, we are only just beginning to elucidate the organization and dynamics of synaptic actin networks formed by primary B cells. Here, we show that LFA-1:ICAM-1 interactions in primary B cells stimulate the formation of a contractile actomyosin arc network that occupies the pSMAC portion of the synapse. This actomyosin network represents the major actin structure at the IS of primary B cells receiving integrin co-stimulation, and its dynamics drive antigen centralization by sweeping antigen centripetally. Importantly, under conditions of limiting antigen, where integrin co-stimulation is required for IS formation, blocking the contractility of this pSMAC network inhibits IS formation and BCR signaling. Finally, we show that germinal center (GC) B cells can also create this actomyosin structure, suggesting that it may contribute to the function of GC B cells as well. Together, our data demonstrate that a contractile actomyosin arc network created downstream of integrin ligation plays a major role in the mechanism by which integrin co-stimulation promotes B-cell activation and IS formation when antigen is limiting. Importantly, these findings highlight the need for including integrin co-stimulation when examining the role of actin during B-cell activation, especially under physiologically relevant conditions.

## Results

### Integrin co-stimulation promotes the formation of an actin arc network in the pSMAC

To investigate the possibility that LFA-1 ligation might also promote B-cell activation by triggering a significant change in synaptic actin organization, we imaged F-actin at ISs formed by primary mouse B cells on glass surfaces coated with either anti-IgM or anti-IgM plus ICAM-1. F-actin was visualized using GFP-F-Tractin, a dynamic reporter for F-actin (*Yi et al., 2012*; *Murugesan et al., 2016*), and two super-resolution imaging modalities: Airyscan (*xy* resolution ~140 nm) and total internal reflection fluorescence-structured illumination microscopy (TIRF-SIM; *xy* resolution ~100 nm). Individual video frames of anti-IgM-engaged B cells using both imaging modalities (*Figure 1A and B*), together with the corresponding videos (*Video 1A and B*), revealed a thin, bright, highly dynamic outer rim of F-actin (white arrows in *Figure 1A and B*) that likely corresponds to the branched actin network comprising the dSMAC (*Bolger-Munro et al., 2019*; *Wang et al., 2018*; *Wang et al., 2017*). Both modalities (but especially TIRF-SIM) showed that the F-actin present inside this outer dSMAC rim is composed of a highly disorganized mixture of short-actin filaments/fibers and actin foci (blue brackets in *Figure 1A and B*), similarly to those observed previously in HeLa cells (*Fritzsche et al., 2017*). In sharp contrast, individual video frames of anti-IgM + ICAM-1-engaged B cells using both modalities (*Figure 1C and D*), together with the corresponding videos (*Video 2A and B*), showed a highly organized network

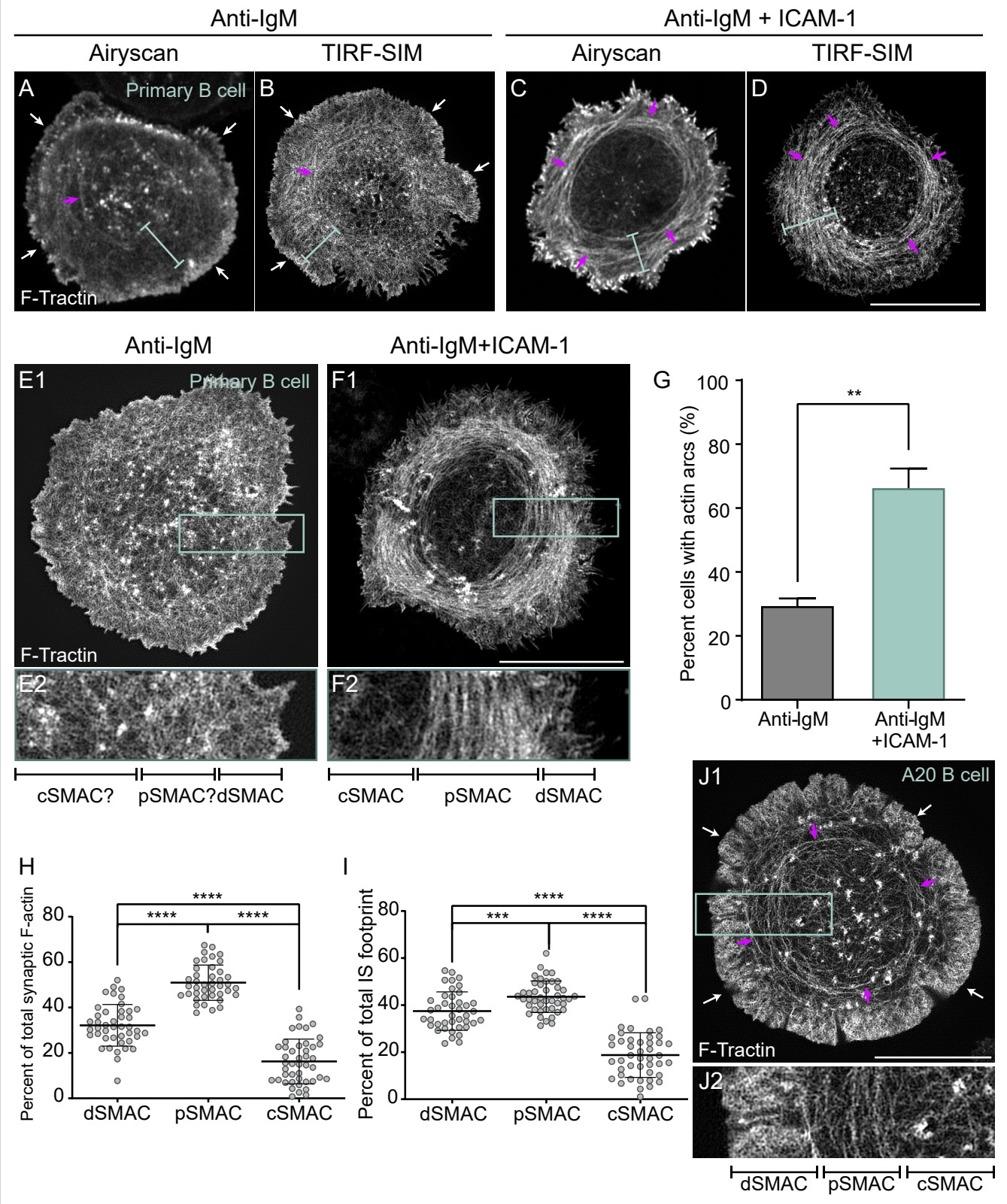

**Figure 1.** ICAM-1 co-stimulation promotes the formation of actin arcs at the B-cell immune synapse. (**A–F**) GFP-F-Tractin-expressing primary B cells on glass coated with anti-IgM alone (**A, B, E1, E2**) or with anti-IgM + ICAM-1 (**C, D, F1, F2**) and imaged using Airyscan (**A, C**) or TIRF-SIM (**B, D, E1, E2, F1, F2**). The white arrows in (**A**) and (**B**) indicate the thin outer rim of dendritic actin in the dSMAC. The blue bars in (**A–D**) indicate the pSMAC. (**E2**) and (**F2**) correspond to the boxed regions in (**E1**) and (**F1**), respectively. Of note, the cell shown in (**E1/E2**) is representative of ~70% of anti-IgM-stimulated

*Figure 1 continued on next page*

*Figure 1 continued*

cells, while the cell shown in (**F1/F2**) is representative of ~70% of anti-IgM + ICAM-1-stimulated cells. (**G**) Percent of cells with pSMAC actin arcs (N > 67 cells/condition from three experiments). (**H, I**) Percent of total synaptic F-actin (**H**) and percent of total IS footprint (**I**) contained within the dSMAC, pSMAC, and cSMAC portions of the synapse for primary B cells on anti-IgG/ICAM-1-coated glass (N = 44 cells/condition from six experiments). (**J1, J2**) GFP-F-Tractin-expressing A20 B cell on anti-IgG/ICAM-1-coated glass. (**J2**) corresponds to the boxed region in (**J1**). The magenta arrows in (**A–D**) and (**J1**) indicate actin arcs. Scale bars: 10 μm.

The online version of this article includes the following figure supplement(s) for figure 1:

**Figure supplement 1.** Degree of alignment between the actin filaments in the pSMAC of B cells stimulated with anti-IgM alone versus anti-IgM and ICAM-1.

inside the outer dSMAC rim (i.e., in the pSMAC) that is comprised of concentric actin arcs (blue brackets and magenta arrows in *Figure 1C and D*). The difference in synaptic actin organization between anti-IgM-engaged B cells and anti-IgM + ICAM-1-engaged B cells is very evident in enlarged TIRF-SIM images. While it is challenging to define SMAC boundaries and any pattern of F-actin organization in the pSMAC of B cells engaged with anti-IgM alone (*Figure 1E1 and E2*), SMAC boundaries and pSMAC F-actin organization are both very distinct in B cells engaged using anti-IgM + ICAM-1 (*Figure 1F1 and F2*). Consistently, scoring B cells for the presence of any discernable arcs showed that the addition of ICAM-1 increases the percentage of such cells from ~30% to ~70% (*Figure 1G*). Importantly, static and dynamic imaging showed that the arcs in cells engaged with anti-IgM alone are sparse and transient (*Figure 1A and B*, *Video 1A and B*), while those in cells engaged with both anti-IgM and ICAM-1 are dense and persistent (*Figure 1C and D*, *Video 2A and B*). In other words, when B cells receiving only anti-IgM stimulation do form discernible arcs (e.g., see those marked by magenta arrows in *Figure 1A and B*), they are much sparser and less persistent than those formed by cells also receiving ICAM-1 stimulation. Moreover, we could not find any B cells receiving anti-IgM stimulation alone that possessed a robust actin arc network. Consistently, measuring the degree of alignment between actin filaments in the pSMAC portion of B cells stimulated with anti-IgM alone versus both anti-IgM and ICAM-1, which were made using FibrilTool (*Boudaoud et al., 2014*), revealed a large shift towards more organized pSMAC actin when ICAM-1 was included (*Figure 1—figure supplement 1A1–A3*; see the figure legend for details). Finally, measuring the percentage of total synaptic F-actin content within each SMAC (*Figure 1H*), and the percentage of total IS footprint occupied by each SMAC (*Figure 1I*), showed that the actin arc-containing pSMAC comprises the major actin network at the IS of primary B cells engaged using both anti-IgM and ICAM-1. Together, these results demonstrate that LFA-1 co-stimulation promotes the formation of a pSMAC actin arc network that dominates the B-cell IS.

## Linear actin filaments generated by the formin mDia1 at the outer edge of the synapse give rise to the pSMAC actin arc network

We next sought to define the origin of the actin arcs that comprise the pSMAC of B cells stimulated using both anti-IgM and ICAM-1. Primary B cells stimulated in this way exhibit small, actin-rich surface spikes at the outer synapse edge (*Figure 2A*).

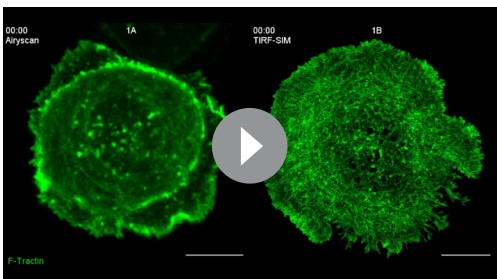

**Video 1.** Representative primary B cells expressing GFP-F-Tractin on glass coated with anti-IgM that were imaged every 3 s for 120 s using Airyscan (A) and TIRF-SIM (B). Played back at 10 fps. Scale bar: 5 μm.

https://elifesciences.org/articles/72805/figures#video1

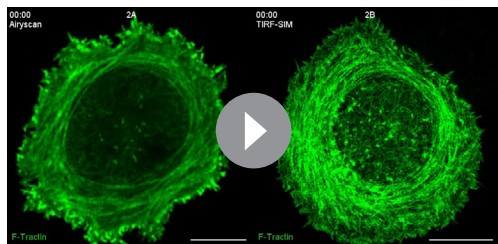

**Video 2.** Representative primary B cells expressing GFP-F-Tractin on glass coated with anti-IgM and ICAM-1 that were imaged every 2 s for 120 s using Airyscan (A) and every 5 s for 600 s using TIRF-SIM (B). Played back at 10 fps. Scale bars: 5 μm.

https://elifesciences.org/articles/72805/figures#video2

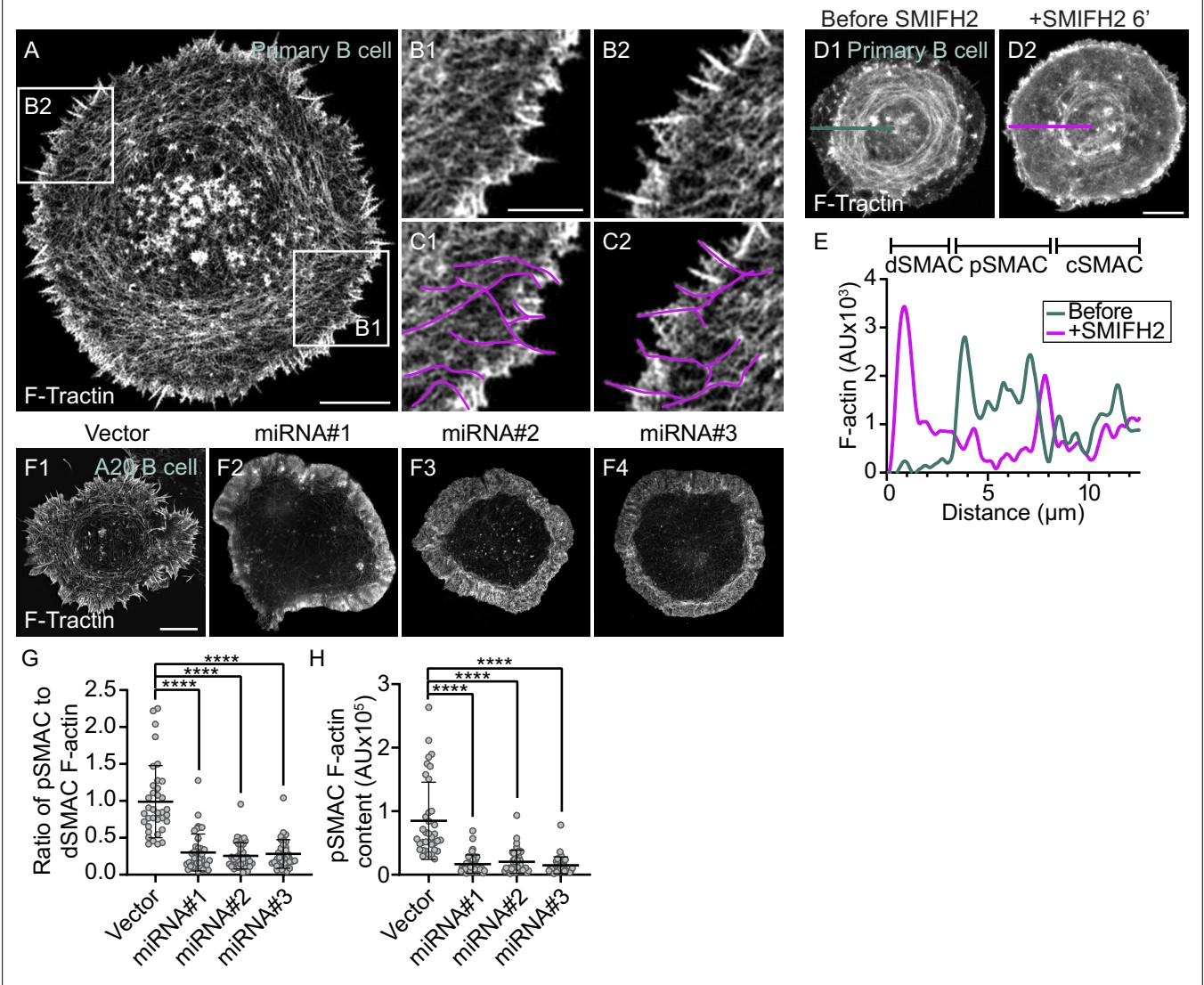

**Figure 2.** The actin arcs are created by the formin mDia1 acting at the outer edge of the immune synapse. (**A**) GFP-F-Tractin-expressing primary B cell on anti-IgG/ICAM-1-coated glass. (**B1, B2**) Boxed regions in (**A**). (**C1, C2**) **B1** and **B2** with magenta lines applied to highlight linear actin filaments/ bundles arising from surface spikes at the IS edge that are contiguous with actin arcs in the pSMAC. (**D1, D2**) GFP-F-Tractin-expressing primary B cell on anti-IgG/ICAM-1-coated glass before (**D1**) and 6 min after SMIFH2 addition (**D2**). (**E**) F-actin intensity profiles corresponding to the line scans in (**D1**) (blue, before SMIFH2 addition) and (**D2**) (magenta, after SMIFH2 addition). (**F1–F4**) F-Tractin mNeonGreen-expressing A20 B cells transfected with vector only or the indicated mDia1 miRNA constructs and activated on anti-IgG/ICAM-1-coated glass. (**G**) Ratio of pSMAC to dSMAC F-actin (N > 20 cells/condition from two experiments). (**H**) pSMAC F-actin content (N = 20–26 cells/condition from two experiments). (**A–C, F**) TIRF-SIM images; (**D**) Airyscan images. Scale bars: 5 µm in (**A, D2, F1**); 2 µm in (**B1**).

The online version of this article includes the following source data and figure supplement(s) for figure 2:

**Figure supplement 1.** miRNA-mediated knockdown (KD) of mouse mDia1 in A20 B cells.

**Figure supplement 1—source data 1.** Western blots of miRNA-mediated knockdown (KD) of mouse mDia1 in A20 B cells.

**Figure supplement 2.** Arp2/3 inhibition shifts the balance between the dSMAC branched actin network and the pSMAC actin arc network.

Importantly, magnified images revealed that the actin within these spikes continues into the cyto-plasm in the form of linear actin filaments (*Figure 2B1 and B2*). Moreover, tracing these linear actin filaments showed that they are contiguous with the pSMAC actin arcs (*Figure 2C1 and C2*, *Video 3A and B*). These results argue that linear actin filaments nucleated at the plasma membrane at the outer edge of the synapse give rise to the actin arcs populating the pSMAC. While these results do not identify the specific nucleator involved, they do point to it being a member of the formin family based

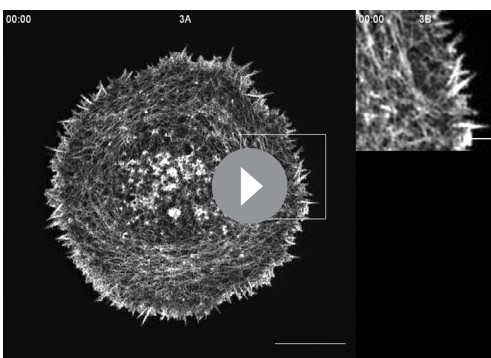

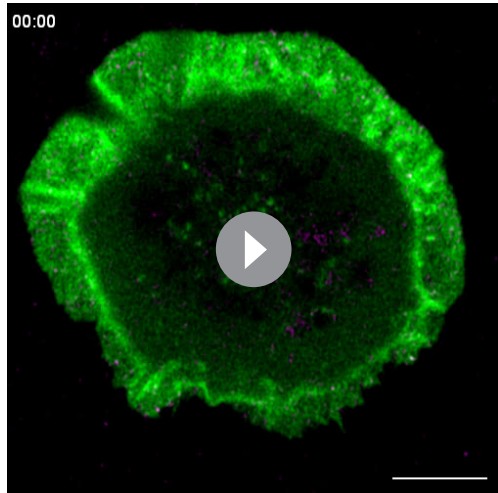

**Video 3.** A representative primary B cell expressing GFP-F-Tractin on glass coated with anti-IgM and ICAM-1 that was imaged every 3 s for 300 s using TIRF-SIM (A). A magnified view of the region boxed in white in (A) is shown in (B).Played back at 10 fps. Scale bars: 5 µm (A), 1 µm (B).

https://elifesciences.org/articles/72805/figures#video3

**Video 4.** A representative A20 B cell expressing mEOS-actin on glass coated with anti-IgG and ICAM-1 that was imaged every 1.8 s for 70 s using Airyscan. Played back at 7 fps. Scale bar: 5 µm.

https://elifesciences.org/articles/72805/figures#video4

on the fact that the actin being made is linear and nucleated at the plasma membrane (*Goode and Eck, 2007*; *Breitsprecher and Goode, 2013*). Consistent with this conjecture, and with the fact that formins incorporate fluorescent protein-labeled actin monomer into filaments poorly (*Yi et al., 2012*; *Murugesan et al., 2016*; *Chen et al., 2012*), we did not see fluorescent actin arcs in B cells expressing mEOS-labeled G-actin (*Video 4*).

To test if a formin is indeed responsible for creating the pSMAC actin arc network, we used the pan-formin inhibitor SMIFH2 (*Rizvi et al., 2009*). *Figure 2D1 and D2*, together with the line scan in *Figure 2E*, show that the pSMAC actin arcs present in a representative primary B cell immediately before SMIFH2 addition (blue trace) had largely disappeared 6 min after adding SMIFH2 (magenta trace). Given recent concerns about the specificity of SMIFH2 (*Nishimura et al., 2021*), we used three different miRNAs to knock down the formin mDia1 in the lymphoma B cell line A20 (*Figure 2—figure supplement 1A and B*), which also forms pSMAC actin arcs when stimulated using anti-IgG + ICAM-1 (*Figure 1J1 and J2*, *Video 5A*). mDia1 was chosen as the miRNA target as it is highly expressed in B cells (ImmGen Database) and is largely responsible for making actin arcs in T cells (*Murugesan et al., 2016*). Compared to control A20 B cells (*Figure 2F1*), representative B cells expressing each of the three miRNAs (*Figure 2F2–F4*) were largely devoid of actin arcs. This difference was supported by quantitating the ratio of pSMAC to dSMAC F-actin (*Figure 2G*), as well as the amount of F-actin in the pSMAC (*Figure 2H*). Finally, actin arcs were unaffected by the expression of two different nontargeting miRNAs (*Figure 2—figure supplement 1C1–C4 and D1–D4*). Together, these results argue that the pSMAC actin arcs are indeed created by a formin, and that the formin mDia1 likely plays a major role.

To provide further evidence that the arcs are created by a formin, we imaged A20 B cells following the addition of the Arp2/3 inhibitor CK-666. The rationale for this experiment lies in the recent revelation that the two major consumers of actin monomer in cells, the Arp2/3 complex and formins, are always competing for a limiting pool of actin monomer (*Burke et al., 2014*; *Lomakin et al., 2015*; *Fritzsche et al., 2016*; *Hammer et al., 2019*). One consequence

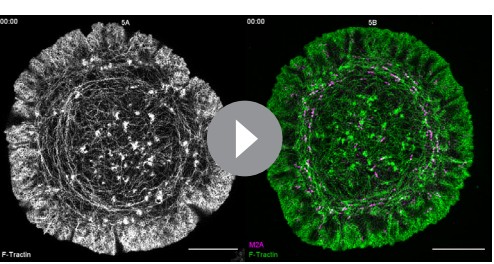

**Video 5.** A representative A20 B cell expressing GFP-F-Tractin on glass coated with anti-IgG and ICAM-1 that was imaged every 1.5 s for 120 s using TIRF-SIM (A). A representative A20 B cell in which we had inserted mScarleti at the N-terminus of M2A using CRISPR (magenta) that was transfected with GFP-F-Tractin (green), activated on glass coated with anti-IgG and ICAM-1, and imaged every 3 s for 120 s using TIRF-SIM (B). Played back at 10 fps. Scale bars: 5 µm.

https://elifesciences.org/articles/72805/figures#video5

of this competition is that when one of these nucleators is inhibited, the actin structures created by the other nucleator get more robust because that nucleator now gets more monomer. For example, inhibiting the Arp2/3 complex promotes the formation of formin-dependent actin networks in both yeast and vertebrate cells (*Murugesan et al., 2016*; *Burke et al., 2014*; *Lomakin et al., 2015*; *Fritzsche et al., 2016*; *Hammer et al., 2019*). Given this, and assuming that the arcs in B cells are formin-generated, then inhibiting the Arp2/3 complex in B cells should lead not only to a diminution of the branched actin network in the dSMAC, but also to an amplification of the arc network in the pSMAC. Consistently, *Figure 2—figure supplement 2A1/A2* (before CK-666 addition) and *Figure 2—figure supplement 2A3/A4* (after CK-666 addition) together show that CK-666 addition leads not only to a reduction in the size of the dSMAC (magenta brackets), but also to an increase in arc content in the pSMAC (blue brackets). These changes were supported by measuring the percentage of total synaptic F-actin content residing within each SMAC (*Figure 2—figure supplement 2B*), which revealed a significant shift away from dSMAC F-actin and toward pSMAC F-actin following CK-666 treatment. This shift was also reflected in measurements of total pSMAC F-actin content (*Figure 2—figure supplement 2C*), the ratio of pSMAC to cSMAC F-actin content (*Figure 2—figure supplement 2D*), and the ratio of pSMAC to cSMAC area (*Figure 2—figure supplement 2E*). Taken together, these data argue strongly that linear actin filaments generated by the formin mDia1 at the outer edge of the synapse give rise to the pSMAC actin arc network.

## Myosin 2A co-localizes with the actin arcs

Having established that ICAM-1 co-stimulation promotes the formin-dependent formation of actin arcs in the pSMAC, we asked how these arcs are organized into concentric structures. Formin-derived linear actin filaments are commonly organized into well-defined structures such as stress fibers, transverse arcs, and the contractile ring in dividing cells by bipolar filaments of the actin-based motor protein myosin 2 (*Vicente-Manzanares et al., 2009*; *Sellers, 2000*; *Shutova and Svitkina, 2018*). We decided, therefore, to test whether myosin 2 co-localizes with the actin arcs and is required for their concentric organization.

To define the localization and dynamics of myosin 2 at the B-cell IS, we used primary B cells isolated from a mouse in which GFP had been knocked into the N- terminus of the myosin 2A heavy chain gene *Myh9* (referred to herein as M2A) (*Zhang et al., 2012*) as M2A is the only myosin 2 isoform expressed in B cells (ImmGen Database). Individual video frames of these cells following transfection with Td-Tomato-F-Tractin and attachment to coverslips coated with anti-IgM and ICAM-1 revealed a dramatic co-localization between M2A and the actin arcs in the pSMAC (*Figure 3A1–A3*, *Video 6*). Magnified TIRF-SIM images show that the myosin signals align with actin arcs in a periodic fashion (*Figure 3A4*) that resembles other myosin 2-rich, linear actin structures like stress fibers and the contractile ring (*Beach et al., 2014*). Moreover, these myosin signals exhibit the SIM signature for M2A bipolar filaments when M2A is GFP-labeled at its N-terminus (*Beach et al., 2017*), which is a pair of GFP puncta spaced ~300 nm apart (*Figure 3A5*; 304 ± 32 nm; n = 230 filaments from 12 cells). The presence of M2A filaments in the medial portion of the synapse was also evident in primary B cells isolated from a mouse in which mCherry had been knocked into the N-terminus of M2A (*Figure 3—figure supplement 1A*), primary B cells that we genome-edited using CRISPR to place GFP at the N-terminus of M2A (*Figure 3—figure supplement 1B*), and A20 B cells that we genome edited using CRISPR to place mScarleti at the N-terminus of M2A and then transfected with GFP-F-Tractin (*Figure 3—figure supplement 1C1–C3*, *Video 5B*). Finally, 3D-SIM images of A20 B cells that were fixed and stained for M2A and F-actin showed that endogenous M2A also co-localizes with the actin arcs (*Figure 3—figure supplement 1D1–D3*; note that the signature for M2A filaments using this antibody, which recognizes the C-terminus of M2A, is a single fluorescent punctum that corresponds to the center of an individual M2A filament) (*Beach et al., 2014*; *Beach and Hammer, 2015*). The extent of this co-localization was even clearer in enlarged images of immunostained cells (*Figure 3—figure supplement 1E1–E3*), where line scans showed endogenous M2A coinciding with actin arcs (*Figure 3—figure supplement 1F*). Together, these results show that the actin arc network in primary B cells receiving ICAM-1 co-stimulation is in fact an actomyosin arc network.

To gain insight into how the arcs become decorated with M2A filaments, we examined time-lapse TIRF-SIM images of GFP-M2A knockin primary B cells expressing Td-Tomato F-Tractin. Individual video frames (*Figure 3B1–B6*), as well as the corresponding video (*Video 7*), show that bipolar filaments of

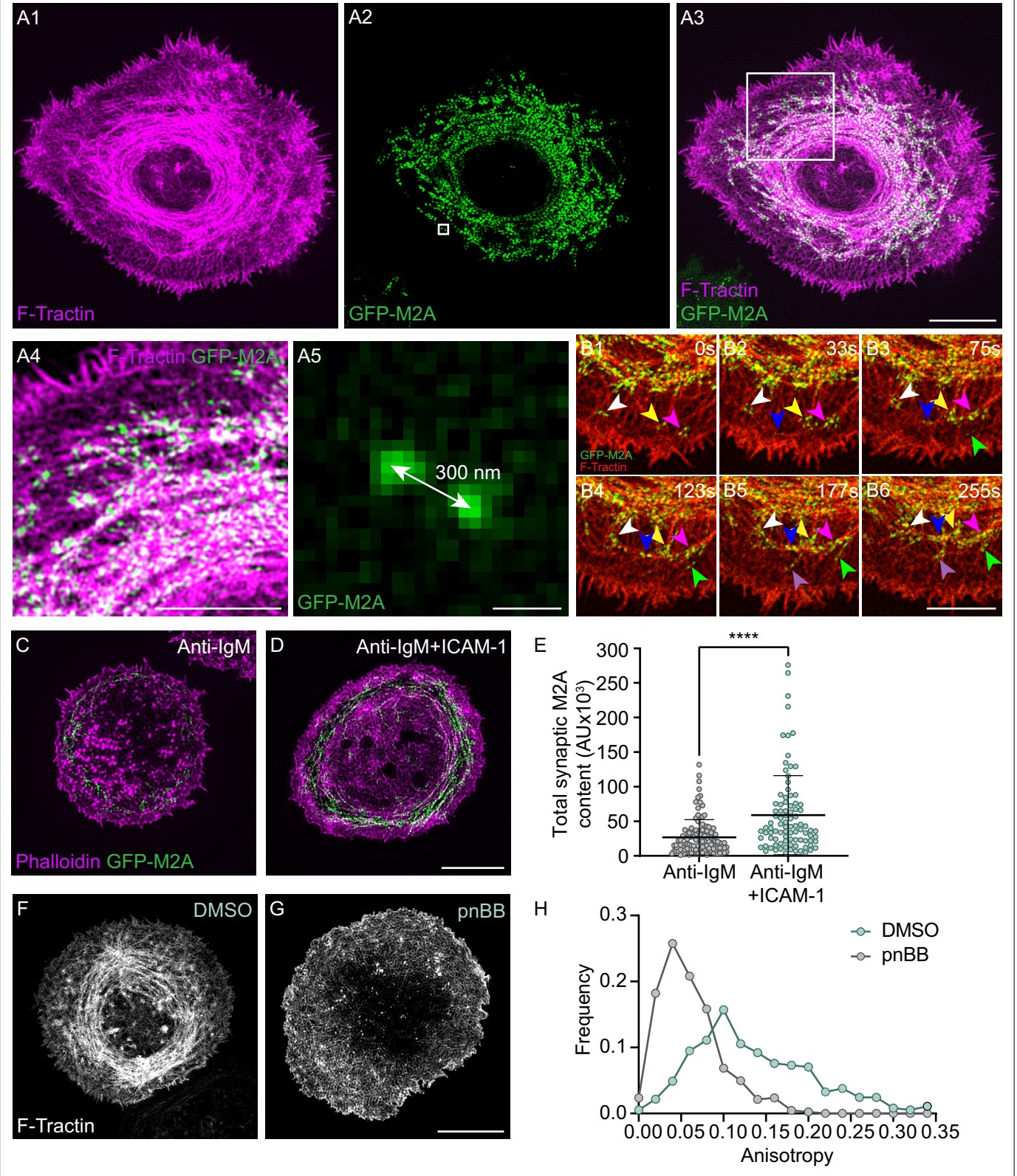

**Figure 3.** Myosin 2A decorates the actin arcs and is required for their concentric organization. (A1–A5) Td-Tomato-F-Tractin-expressing primary B cell from the M2A-GFP knockin mouse on anti-IgM/ICAM-1-coated glass. (A4) and (A5) correspond to the boxed regions in (A1) and (A2), respectively. (B1–B6) Still images at the indicated time points taken from a region within *Video 7* of a Td-Tomato-F-Tractin-expressing primary B cell from the M2A-GFP knockin mouse. Different color arrowheads mark the formation and centripetal movement of individual M2A bipolar filaments (see text for details). (C,

*Figure 3 continued on next page*

*Figure 3 continued*

**D**) Phalloidin-stained primary B cell from the M2A-GFP knockin mouse on glass coated with anti-IgM alone (**C**) or with anti-IgM + ICAM-1 (**D**). (**E**) Total synaptic M2A content (N = 91–115 cells/condition from three experiments). (**F, G**) GFP-F-Tractin-expressing primary B cells that had been pretreated with DMSO (**F**) or pnBB (**G**) for 30 min and activated on anti-IgM/ICAM-1-coated glass. (**H**) Anisotropy of the actin filaments/bundles present within the pSMAC (N = 369–423 regions of interest [ROIs] from 30 to 37 cells from three experiments). All panels: TIRF-SIM images. Scale bars: 5 µm in (**A3, D, G**); 3 µm in (**A4, B6**); 250 nm in (**A5**).

The online version of this article includes the following figure supplement(s) for figure 3:

**Figure supplement 1.** Endogenous M2A decorates the actin arcs in both primary B cells and A20 B cells.

**Figure supplement 2.** Integrin-dependent traction forces exerted by primary B cells require M2A contractility.

M2A begin to appear near the dSMAC:pSMAC boundary in association with the linear actin filaments/bundles exiting the dSMAC (white, yellow, and fuchsia arrowheads mark such myosin filaments at time 0 s in *Figure 3B1*). As time progresses, these filaments move centripetally and undergo expansion into filament clusters (*Figure 3B1–B6*; see also *Video 7*). This expansion, in which individual myosin filaments expand into a small cluster of filaments, is presumably driven by the same sequential amplification pathway described previously for M2A filament assembly in fibroblasts (*Beach et al., 2017*). Finally, the myosin filaments in these clusters begin to align with the arcs forming at the outer edge of the pSMAC, which then merge with the larger actomyosin arc network in the pSMAC (*Figure 3B1–B6*). As all this is happening, new myosin filaments keep appearing near the dSMAC:pSMAC boundary to repeat the process (*Figure 3B2–B6*; follow the blue, green, and purple arrowheads).

Given that ICAM-1 co-stimulation promotes the formation of actin arcs and that the arcs recruit M2A, ICAM-1 co-stimulation should also result in an increase in the amount of M2A at the IS. Consistently, primary GFP-M2A knockin B cells receiving both anti-IgM and ICAM-1 stimulation exhibited a greater amount of synaptic M2A than B cells receiving only anti-IgM stimulation (*Figure 3C–E*). Of note, this difference remained significant even after normalizing the M2A fluorescence for a small difference in the average cell-spread area under these two conditions (*Figure 3—figure supplement 1G1–G2*).

## Myosin 2A contractility is required for the concentric organization of the actin arcs and integrin-dependent traction force

The organization of formin-generated linear actin filaments into well-defined structures is typically driven by the contractility of myosin 2 filaments (*Vicente-Manzanares et al., 2009*; *Sellers, 2000*). There-

fore, we asked if M2A contractility is required for the concentric organization of the pSMAC actin arcs by treating cells with para-nitroblebbistatin (pnBB), a blue light-insensitive version of the cell-permeable, small molecule myosin 2 inhibitor blebbistatin (BB) that blocks myosin 2-based contractility by locking the myosin in its weak actin binding state (*Képiró et al., 2014*). While control, DMSO-treated cells exhibited concentric actin arcs in their pSMAC as expected (*Figure 3F*), cells

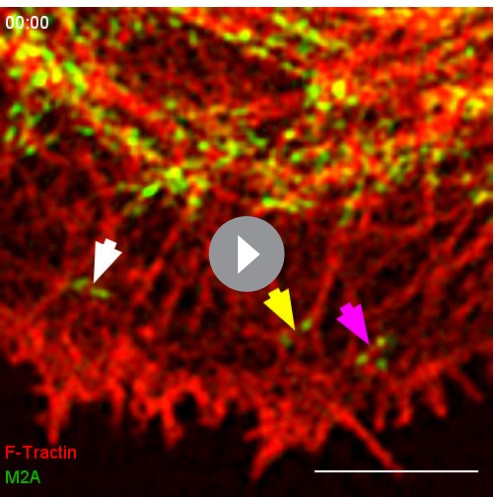

**Video 7.** A magnified view of a region within *Video 6*. The applied arrowheads mark various aspects of M2A filament assembly and organization as explained in the text for Figure 3B1–B6. Scale bar: 1 µm.
https://elifesciences.org/articles/72805/figures#video7

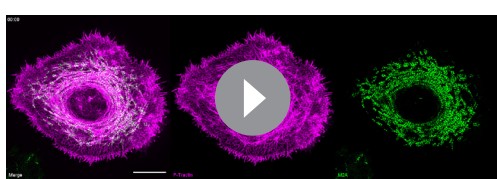

**Video 6.** A representative primary B cell from a M2A-GFP knockin mouse expressing Td-Tomato-F-Tractin on glass coated with anti-IgM and ICAM-1 that was imaged every 3 s for 300 s using TIRF-SIM. Played back at 10 fps. Scale bar: 5 µm.
https://elifesciences.org/articles/72805/figures#video6

treated with 25 µM pnBB displayed highly disorganized, mesh-like actin arrays in their pSMAC (*Figure 3G*). Consistently, anisotropy measurements made using FibrilTool revealed a dramatic shift towards more disorganized pSMAC actin when B cells are treated with pnBB (*Figure 3H*). Together, these results demonstrate that M2A contractility is indeed required for the concentric organization of the pSMAC actin arcs.

We used traction force microscopy in combination with pnBB to ask if integrin-dependent traction forces that B cells exert on a deformable substrate require M2A contractility. As expected (*Wang, 2018*; *Kumari et al., 2019*), B cells engaged with substrate coated with anti-IgM and ICAM-1 generated significantly more traction force than B cells engaged with substrate coated with anti-IgM alone (*Figure 3—figure supplement 2A1, A2, B1, B2, and D*). Importantly, ICAM-1-dependent traction forces were completely abrogated by pretreating the cells with pnBB (*Figure 3—figure supplement 2C1, C2, and D*), indicating that the generation of integrin-dependent traction forces requires M2A contractility. This requirement likely reflects pulling forces exerted by M2A on the substrate through LFA-1:ICAM-1 pairs, combined with the increase in M2A content at the synapse caused by ligating LFA-1 with ICAM-1, and the contribution that M2A-dependent pulling forces make in keeping LFA-1 in its open, active conformation (*Gardel et al., 2010*; *Case and Waterman, 2015*; *Comrie and Burkhardt, 2016*). These results, together with the fact that integrin clusters are known to accumulate in the pSMAC portion of the B-cell IS (*Carrasco et al., 2004*; *Carrasco and Batista, 2006b*), suggest a feed-forward relationship where integrin ligation promotes the formation of pSMAC actomyosin arcs, and the contractile forces exerted by these actomyosin arcs promote further integrin activation and robust adhesion in the pSMAC.

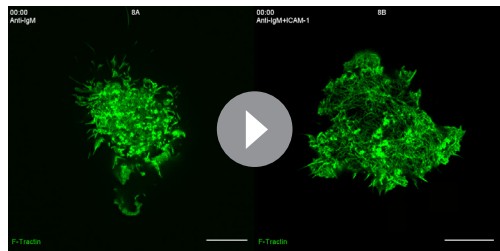

**Video 8.** Representative primary B cells expressing GFP-F-Tractin on PLBs containing anti-IgM and ICAM-1 (A) or anti-IgM alone (B) that were imaged every 5 s for 300 s using TIRF-SIM and played back at 8 fps. Scale bar: 5 µm.

https://elifesciences.org/articles/72805/figures#video8

## The actomyosin arc network in the pSMAC exhibits centripetal flow

Inward flows of cortical actin networks are thought to drive the transport of antigen receptor clusters to the center of maturing synapses in both T cells and B cells (*Wang and Hammer, 2020*; *Hammer et al., 2019*; *Blumenthal and Burkhardt, 2020*; although see *Schnyder et al., 2011*; *Babich and Burkhardt, 2011*). For B cells, the clearest example of this to date is the demonstration that the centripetal flow of the branched actin network comprising the dSMAC propels BCR:antigen clusters towards the cSMAC (*Bolger-Munro et al., 2019*). As a prelude to asking whether the actomyosin arcs comprising the pSMAC also contribute to antigen centralization, we asked if this contractile network exhibits centripetal flow. Kymograph analyses of actin flow across synapses made by primary B cells expressing GFP-F-Tractin showed that their pSMAC actomyosin arc network indeed flows centripetally at 1.07 ± 0.07 µm/min, or about one-third the rate of centripetal actin flow in the dSMAC (2.89 ± 0.18 µm/min) (*Figure 4—figure supplement 1A1–A3*). Similar results were obtained for A20 B cells (pSMAC rate: 0.97 ± 0.13 µm/min; dSMAC rate: 3.16 ± 0.35 µm/min) (*Figure 4—figure supplement 1B1–B3*). Together, these results indicate that the actomyosin arcs could contribute along with the branched actin network in the dSMAC to the inward transport of BCR:antigen clusters.

## Actomyosin arcs contribute to antigen centralization by sweeping BCR:antigen clusters inward

We used PLBs to determine if the actomyosin arcs do in fact contribute to antigen centralization. As expected, primary B cells expressing GFP-F-Tractin readily formed actin arcs when PLBs contained both anti-IgM and ICAM-1 (*Video 8A*), but not when they contained anti-IgM alone (*Video 8B*). Also as expected, primary B cells engaged with PLBs containing fluorescent anti-IgM (magenta) and unlabeled ICAM-1 yielded mature synapses in which concentric actin arcs surrounded antigen accumulated in the cSMAC (*Figure 4A1–A3*, white arrows). To obtain a holistic view of antigen centralization, we imaged antigen clusters in the dSMAC and pSMAC of primary B cells over time with the aim of

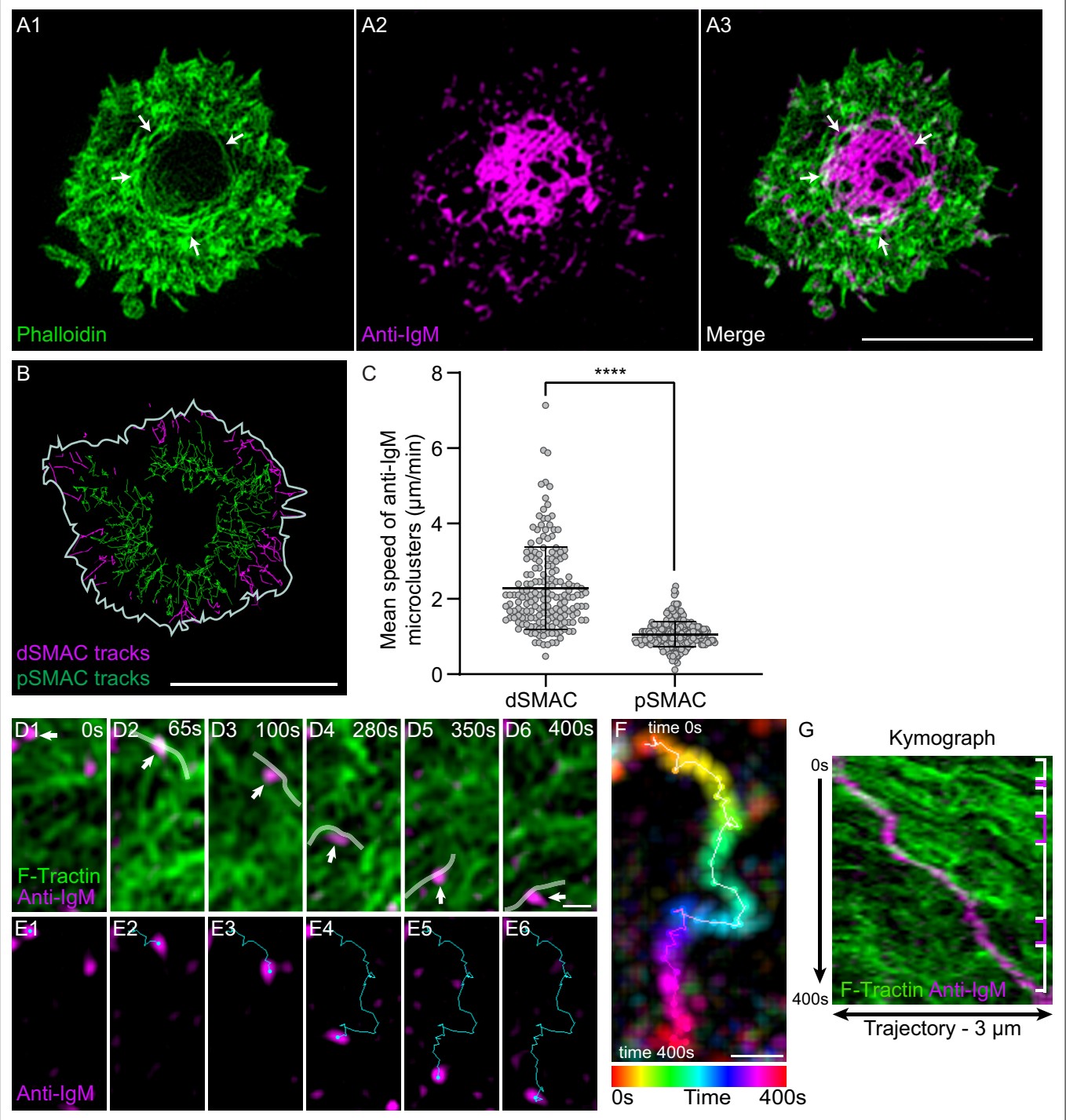

**Figure 4.** Actin arcs sweep antigen clusters centripetally. (**A1–A3**) Phalloidin-stained (green) primary B cell 15 min after engagement with a PLB containing unlabeled ICAM-1 and limiting anti-IgM (magenta). The white arrows in (**A1**) and (**A3**) mark the actin arcs. (**B**) Tracks of single anti-IgM microclusters traveling centripetally across the dSMAC (magenta tracks) and pSMAC (green tracks) acquired from *Video 9*. The white line indicates the outer edge of this cell. (**C**) Mean speed of single anti-IgM microclusters moving centripetally across the dSMAC and pSMAC (N = 180–273 tracks from three well-spread cells). (**D1–D6**) Still images at the indicated time points from *Video 10* showing the centripetal movement of actin arcs and a representative anti-IgM microcluster (white arrows) (the center of the synapse is directly below the images). Transparent white lines highlight the actin arcs that moved the microcluster centripetally. (**E1–E6**) Same as (**D1–D6**) except showing only the anti-IgM microcluster and indicating its centripetal path in blue. (**F**) Temporally pseudo-colored, projected image of the anti-IgM microcluster in (**D**) and (**E**). (**G**) Kymograph of the 3-μm-long paths taken by the microcluster and the actin arcs in (**D**) and (**E**) over a period of 400 s. The white brackets on the right indicate where actin arcs overlapped with

*Figure 4 continued on next page*

*Figure 4 continued*

and moved the microcluster, while the magenta brackets indicate where the movement of the microcluster stalled. (**A**) Airyscan images; (**D–G**) TIRF-SIM images. Scale bars: 5 µm in (**A3, B**); 300 nm in (**D6, F**).

The online version of this article includes the following figure supplement(s) for figure 4:

**Figure supplement 1.** Centripetal actin flow rates across the dSMAC and pSMAC portions of synapses made by primary B cells and A20 B cells.

correlating their rates of centripetal transport with the distinct rates of centripetal actin flow exhibited by these two IS zones (***Video 9***). Tracking of single-antigen microclusters showed that they moved inward at 2.36 ± 1.1 µm/min and 1.03 ± 0.3 µm/min across the dSMAC (magenta tracks) and pSMAC (green tracks), respectively (***Figure 4B and C***). Importantly, these rates are very similar to the rates of centripetal actin flow across the dSMAC and pSMAC, respectively (***Figure 4—figure supplement 1A1–A3***). Together, these observations argue that the pSMAC actomyosin arc network works together with the dSMAC branched actin network to drive antigen centralization.

To identify the mechanism by which the actomyosin arcs drive antigen centralization, we imaged F-actin and anti-IgM in the medial portion of forming synapses at high magnification using TIRF-SIM. Anti-IgM microclusters were seen to move across the pSMAC towards the cSMAC (which in the following images was in the down direction) while embedded in an arc network moving in the same direction (***Video 10***). White lines in ***Video 10*** and in the corresponding still images in ***Figure 4D1–D6*** mark actin arcs that were sweeping an individual anti-IgM microcluster inward (***Figure 4E1–E6***). ***Figure 4F*** shows the trajectory of this microcluster (temporally color-coded) as it moved towards the cSMAC. Finally, a kymograph of this trajectory (***Figure 4G***) shows that several actin arcs contributed to the inward movement of this microcluster (areas bracketed in white), and that pauses in movement (areas bracketed in pink) occurred where no actin signal was immediately adjacent to the microcluster. Together, these results argue that individual actin arcs move individual BCR:antigen microclusters inward via a sweeping mechanism that likely depends on frictional coupling between the actin arc and microcluster (***Yu et al., 2010***; ***Ditlev et al., 2019***; ***Smoligovets et al., 2012***). While arcs can slip past microclusters, the overall incidence of such slippage must be fairly small as the rate of inward antigen transport across the pSMAC (***Figure 4C***)

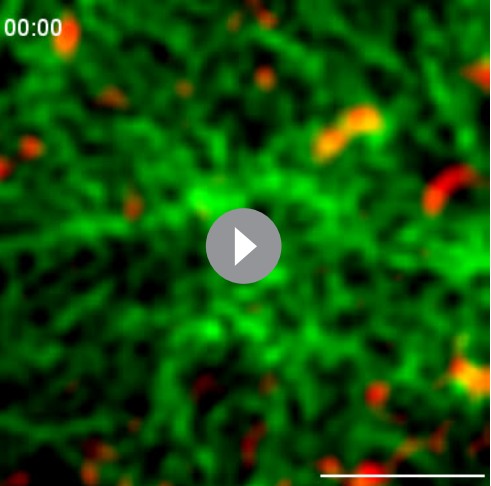

**Video 10.** A region within the pSMAC of a representative primary B cell expressing GFP-F-Tractin (green), engaged with a PLB containing fluorescent anti-IgM (red) and unlabeled ICAM-1, and imaged every 5 s for 400 s using TIRF-SIM. The applied white lines mark actin arcs that are sweeping an antigen cluster centripetally, as explained in the text for Figure 4D1–D6. The inward tracks of this cluster and two other clusters are then shown in blue, green, and red, respectively. Played back at 10 fps. Scale bar: 1 µm.
https://elifesciences.org/articles/72805/figures#video10

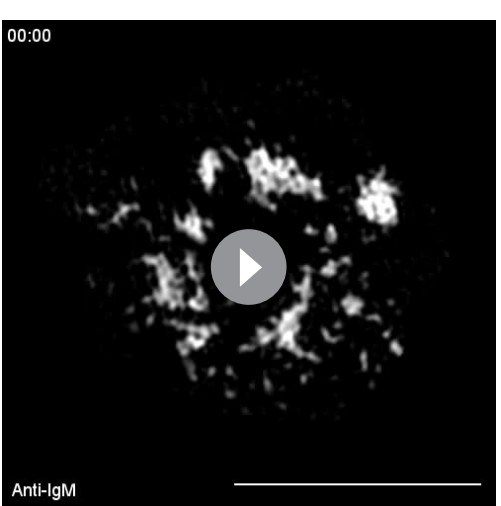

**Video 9.** A representative primary B cell on a PLB containing fluorescent anti-IgM (white) and unlabeled ICAM-1 that was imaged every 5 s for 345 s using TIRF-SIM. Played back at 10 fps. Scale bar: 5 µm.
https://elifesciences.org/articles/72805/figures#video9

is not significantly slower than the rate of inward actin arc flow across the pSMAC (*Figure 4—figure supplement 1A3*).

## Integrin ligation-dependent IS formation requires myosin 2A contractility

B cells engaged with membrane-bound antigen at low density fail to centralize antigen unless their integrin LFA-1 is also engaged with ICAM-1 in the target membrane (*Carrasco et al., 2004*). As a prelude to investigating the myosin dependence of this integrin co-stimulatory effect, we sought to recapitulate these findings using primary B cells and PLBs containing varying amounts of mobile, fluorophore-labeled anti-IgM antibody in the presence or absence of unlabeled ICAM-1. Using this approach, we determined an amount of anti-IgM antibody that would not elicit robust antigen centralization in the absence of ICAM-1, but would in its presence. B cells exhibited robust antigen centralization/cSMAC formation over 10 min without the need for ICAM-1 when the PLB was loaded using a solution containing anti-IgM at a concentration of 2 µg/ml (hereafter referred to as 'high-density antigen') (*Figure 5—figure supplement 1A1–A3*). By contrast, B cells formed antigen microclusters across their synaptic interface but failed to centralize them over 10 min when the PLB was loaded using a solution containing anti-IgM at a concentration 0.15 µg/ml (hereafter referred to as 'low or limiting density antigen') (*Figure 5—figure supplement 1B1–B3*). Importantly, when unlabeled ICAM-1 was included in these low-density antigen bilayers, B cells now exhibited robust antigen centralization/cSMAC formation (*Figure 5—figure supplement 1C1–C3*). This co-stimulatory effect was supported by scoring antigen distribution as centralized, partially centralized or noncentralized (*Figure 5—figure supplement 1D1–D3 and E*). It was also supported by scoring the percent of total synaptic antigen present within the cSMAC, which was defined by a circular area encompassing 20% of the entire synaptic interface and centered around the center of mass of the fluorescent antigen-containing pixels within the interface (*Figure 5—figure supplement 1F*). Finally, it was supported by measuring the size of antigen clusters as a function of their distance from the center of the cSMAC (defined as above) (*Figure 5—figure supplement 1G*). Specifically, B cells engaged with PLBs containing antigen at the limiting density and no ICAM-1 exhibited small antigen clusters (~0.3 µm$^2$) located roughly evenly across the synaptic interface (*Figure 5—figure supplement 1G*, black trace), while B cells engaged with PLBs containing ICAM-1 in addition to antigen at the limiting density exhibited large antigen clusters (up to 3 µm$^2$), the largest of which were located at the center of the cSMAC (*Figure 5—figure supplement 1G*, green trace). Of note, the total amount of antigen present at the synaptic interface was also greater for cells engaged with low-density anti-IgM + ICAM-1 than for cells engaged with low-density anti-IgM alone (*Figure 5—figure supplement 1H*). Together, these results recapitulated a central aspect of the integrin co-stimulatory effect described by *Carrasco et al., 2004*, and they established the specific conditions we used next to test the myosin dependence of this co-stimulatory effect.

To score the myosin dependence of the integrin co-stimulatory effect, we measured the ability of primary B cells treated with either vehicle control (DMSO) or pnBB to centralize antigen and form a cSMAC when engaged for 10 min with PLBs containing ICAM-1 and anti-IgM at the limiting density. While DMSO-treated cells exhibited robust antigen centralization/cSMAC formation (*Figure 5A1–A3*), pnBB-treated cells failed to centralize antigen/create a cSMAC (*Figure 5B1–B3*). Consistently, the actin arcs that surround centralized antigen in DMSO-treated cells (*Figure 5C1–C3*, white arrows) were absent in pnBB-treated cells (*Figure 5D1–D3*). The fact that myosin inhibition abrogates the integrin co-stimulatory effect was further supported by scoring antigen distribution in control and pnBB-treated cells as centralized, partially centralized, or noncentralized (*Figure 5E*), byscoring the percent of total synaptic antigen present within the cSMAC (*Figure 5F*), and bymeasuring the size of antigen clusters as a function of their distance from the center of the cSMAC (*Figure 5G*). Of note, the total amount of antigen present at the synaptic interface was also greater for cells treated with DMSO than for cells treated with pnBB (*Figure 5H*). Together, these results show that the ability of integrin ligation to promote antigen centralization and cSMAC formation when antigen is limiting requires myosin contractility. This in turn argues that the contractile actomyosin arc network created downstream of integrin ligation plays an important role in the mechanism by which LFA-1 co-stimulation promotes B-cell activation.

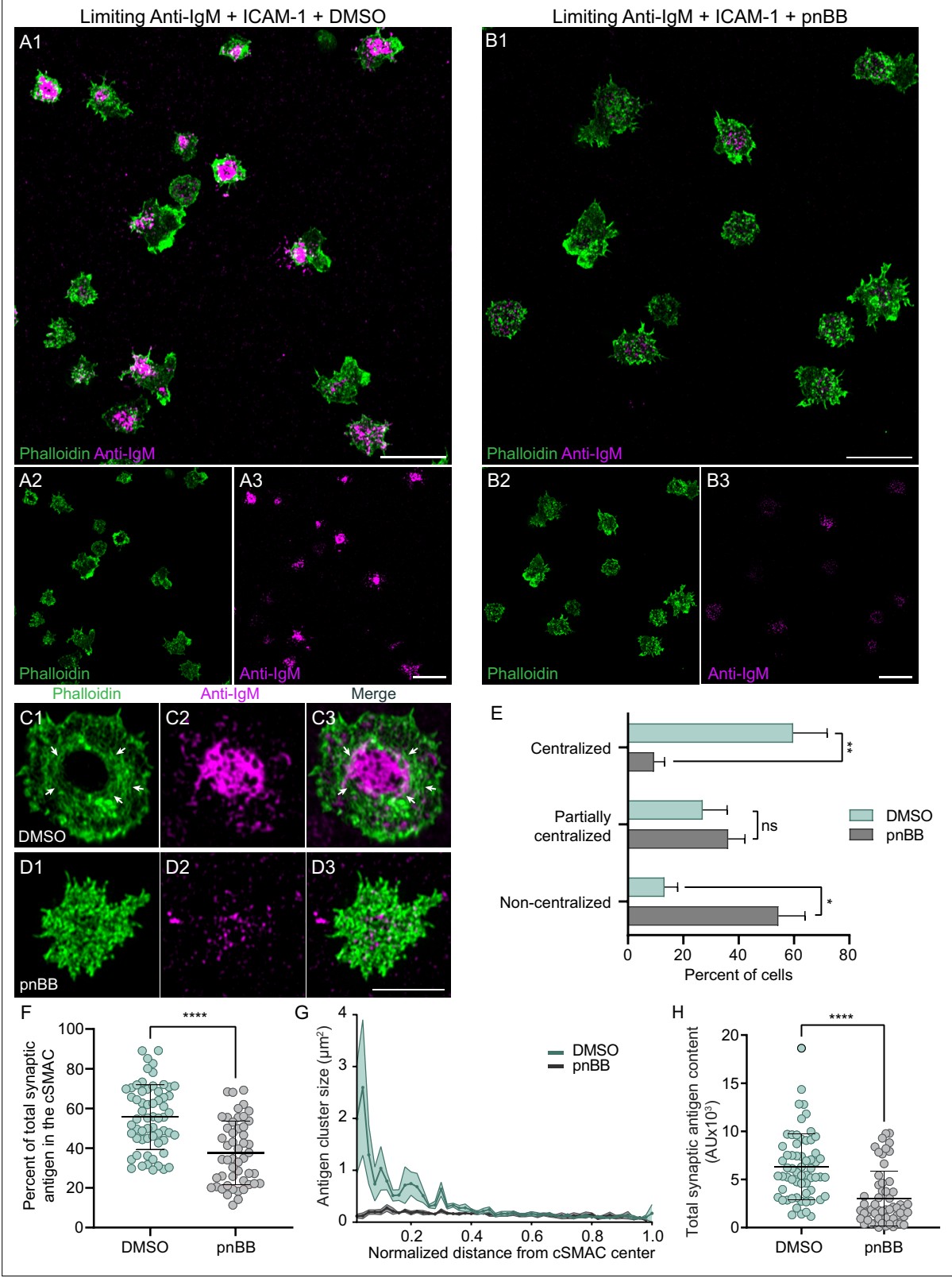

**Figure 5.** Integrin ligation-dependent immune synapse (IS) formation requires myosin 2A contractility. (**A1–A3**) DMSO-treated, phalloidin-stained primary B cells 15 min after engagement with a PLB containing ICAM-1 and limiting anti-IgM. (**B1–B3**) Same as (**A1–A3**) except the B cells were treated with pnBB. (**C1–C3**) Images of a representative, DMSO-treated primary B cell (white arrows mark actin arcs). (**D1–D3**) Images of a representative, pnBB-treated primary B cell. (**E**) Percent of cells exhibiting centralized, partially centralized, and noncentralized antigen (see *Figure 5—figure supplement*

*Figure 5 continued*

*1D1–D3* for representative examples of these three types of antigen distribution) (N = 126–144 cells/condition from three experiments). (**F**) Percent of total synaptic antigen in the cSMAC (N = 81–86 cells/condition from three experiments). (**G**) Antigen cluster size as a function of normalized distance from the cSMAC center (N = 113–144 cells/condition from three experiments). (**H**) Total synaptic antigen content (N = 56–62 cells/condition from three experiments). All panels: Airyscan images. Scale bars: 10 µm in (**A1, B1, A3, B3**); 5 µm in (**D3**).

The online version of this article includes the following figure supplement(s) for figure 5:

**Figure supplement 1.** ICAM-1 co-stimulation promotes antigen centralization and immune synapse (IS) formation when antigen is limiting.

**Figure supplement 2.** M2A contractility potentiates antigen centralization even when antigen density is high.

Finally, we were curious if the robust centralization of antigen that occurs in the absence of LFA-1 ligation when the density of antigen is high is also dependent on myosin contractility, at least to some extent. Indeed, we found that treatment with para-amino BB (paBB), a newer, slightly more water-soluble version of BB (*Várkuti et al., 2016*), attenuated antigen centralization significantly even when the density of antigen was high (*Figure 5—figure supplement 2*; see legend for details), although the magnitude of the inhibition was smaller than for B cells engaged with limiting antigen plus ICAM-1 (compare the results in *Figure 5—figure supplement 2* to the results in *Figure 5*). We conclude, therefore, that M2A contractility potentiates antigen centralization when antigen density is high as well as when antigen density is low enough that LFA-1 co-stimulation becomes important for IS formation. That said, additional experiments should help define how myosin contributes to antigen centralization in B cells receiving only strong anti-IgM stimulation.

## Myosin 2A contractility promotes BCR-dependent signaling

To measure the contribution that actomyosin arcs might make to BCR-dependent signaling, we determined the effect that pnBB has on the distribution and synaptic content of phosphorylated CD79a (P-CD79a), an early signaling molecule responsible for signal transduction downstream of BCR-antigen interaction (*Batista et al., 2010*; *Tanaka and Baba, 2020*). Consistent with results above and with the known properties of CD79a, DMSO-treated primary B cells engaged for 10 min with PLBs containing ICAM-1 and limiting antigen and then fixed/stained for P-CD79a exhibited robust cSMAC formation, with P-CD79a and anti-IgM concentrated in the cSMAC (*Figure 6A1–A4*). Also as expected, pnBB-treated B cells failed to form a clear cSMAC, resulting in CD79a and anti-IgM spread across the synapse (*Figure 6B1–B4*). Importantly, quantitation showed that pnBB-treated cells also exhibited a significant reduction relative to control cells in synaptic P-CD79a content (*Figure 6C*). This defect was also seen after only 5 min on PLBs (*Figure 6—figure supplement 1A*), and the defects at both time points were not due to differences between BB-treated cells and control cells in synaptic CD79a content (*Figure 6—figure supplement 1B*).

To extend these results, we determined the effect that pnBB has on the distribution and synaptic content of phosphorylated CD19, an important co-receptor for the BCR that is responsible for PI3K activation (*Harwood and Batista, 2011*; *Tuveson et al., 1993*; *Keppler et al., 2015*; *Depoil et al., 2008*). DMSO-treated primary B cells engaged with PLBs as above exhibited robust cSMAC formation, with P-CD19 enriched at the outer edge of the IgM concentrated in the cSMAC (*Figure 6D1–D4*). This enrichment of P-CD19 at the pSMAC/cSMAC boundary was confirmed by line scans of the fluorescence intensities for F-actin, anti-IgM, and P-CD19 (*Figure 6G*, see boxed pSMAC regions). In contrast to control cells, pnBB-treated B cells failed to concentrate anti-IgM at the center of the synapse, and P-CD19 staining was now spread across the synaptic interface (*Figure 6E1–E4 and H*). Moreover, quantitation showed that pnBB-treated cells also exhibited a significant reduction relative to control cells in synaptic P-CD19 content (*Figure 6F*) that was not due to a difference in synaptic CD19 content (*Figure 6—figure supplement 1C*). Together, these results indicate that the actomyosin arcs promote BCR-dependent signaling.

## Germinal center B cells can make actomyosin arcs and centralize antigen

Recent studies have presented evidence that GC B cells differ markedly from naïve B cells with regard to the distribution and fate of antigen at mature synapses. Rather than concentrating antigen at the center of the synapse and using actomyosin force to extract it there, GC B cells accumulate antigen in

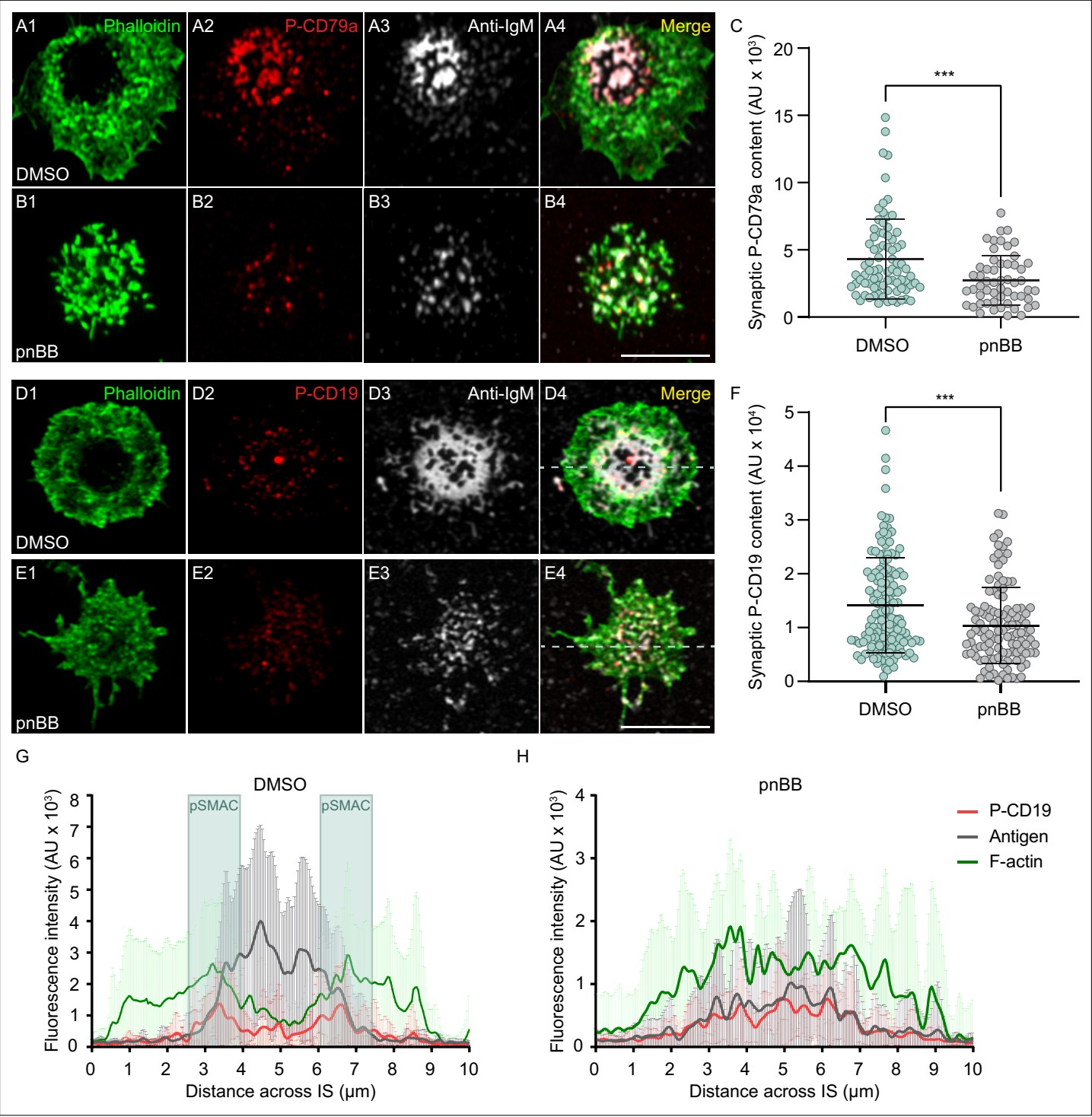

**Figure 6.** Myosin 2A contractility promotes B-cell receptor (BCR) signaling. (**A1–A4**) DMSO-treated primary B cell 10 min after engagement with a PLB containing ICAM-1 and limiting anti-IgM, and stained for F-actin and P-CD79a. (**B1–B4**) Same as (**A1–A4**) except the B cell was treated with pnBB. (**C**) Synaptic P-CD79a content (N = 55–81 cells/condition from three experiments). (**D1–D4**) DMSO-treated primary B cell 10 min after engagement with a PLB containing ICAM-1 and limiting anti-IgM, and stained for F-actin and P-CD19. (**E1–E4**) Same as (**D1–D4**) except the cell was treated with pnBB. (**F**) Synaptic P-CD19 content (N = 115–140 cells/condition from three experiments). (**G**) Fluorescence intensities across synapses for P-CD19 (red), antigen (gray), and F-actin (green) in B cells treated with DMSO (N = 22 cells from two experiments). The position of the pSMAC is highlighted in blue. (**H**) Same as (**G**) except the cells were treated with pnBB (N = 16 cells from two experiments). All panels: Airyscan images. Scale bars: 5 µm in (**B4**); 3 µm in (**E4**).

*Figure 6 continued on next page*

*Figure 6 continued*

The online version of this article includes the following figure supplement(s) for figure 6:

**Figure supplement 1.** Myosin 2A contractility promotes B-cell receptor (BCR) signaling.

clusters at the periphery of the synapse and use actomyosin force to extract it there (*Hammer et al., 2019*; *Nowosad et al., 2016*; *Kwak et al., 2018*). These and other results argue that GC B cells differ dramatically from naïve B cells with regard to the organization of actomyosin at their synapse. We wondered, however, if actomyosin arcs could be detected in mouse GC B cells using our imaging approaches. Consistently, TIRF-SIM imaging of mouse GC B cells isolated from the GFP-M2A knockin mouse that were stained with CellMask Deep Red and plated on coverslips coated with anti-IgM, anti-IgG, and ICAM-1 revealed a subset of cells exhibiting enrichment of M2A filaments in the medial, pSMAC portion of the synapse (*Video 11*), just as in naïve B cells. Moreover, these myosin filaments move centripetally (*Video 11*) and co-localize with pSMAC actin arcs in phalloidin-stained samples (*Figure 7A1–A3*, white arrows), just as in naïve B cells. Importantly, scoring showed that about one-third of GC B cells exhibited robust accumulation of M2A filaments in the pSMAC when engaged with anti-IgM/IgG-coated glass (*Figure 7B*). Similarly, about one-third of GC B cells engaged for 10 min with PLBs containing fluorophore-labeled anti-IgM/IgG and unlabeled ICAM-1, and then fixed and stained with phalloidin, exhibited robust accumulation of M2A filaments in the pSMAC (*Figure 7C1–C4 and D,*). Importantly, these actomyosin arcs can be seen to surround antigen accumulated at the center of the synapse (see the white arrows in *Figure 7C1, C2, and C4*). This finding, together with the fact that the myosin moves centripetally during IS formation (*Video 11*), suggests that actomyosin arcs can contribute to antigen centralization in GC B cells as well as in naïve B cells.

Given these results, we asked if our PLB-engaged mouse GC B cells can centralize antigen. In partial agreement with previous findings (*Nowosad et al., 2016*; *Kwak et al., 2018*), ~45% of synapses exhibited small to medium-sized antigen clusters distributed to varying degrees in the synapse periphery (*Figure 7E1, E2, and F*). In addition, ~20% of synapses exhibited antigen microclusters spread throughout the synaptic interface (*Figure 7E3 and F*). Importantly, the remaining ~35% of synapses exhibited highly centralized antigen (*Figure 7E4 and F*). Images of these synapses showed a pSMAC-like accumulation of GFP-M2A surrounding much of the centralized antigen (*Figure 7—figure supplement 1A1*; see also *Video 12*). Conversely, images of synapses containing either peripheral antigen clusters or microclusters showed no obvious pattern to the distribution of GFP-M2A (*Figure 7—figure supplement 1A2 and A3*; see also *Video 13*). Moreover, the synapses containing peripheral antigen clusters exhibited less total GFP-M2A signal than the synapses with centralized antigen (*Figure 7—figure supplement 1B*), and the signals appeared more transient (compare *Video 13* to *Video 12*). These results, together with the images in *Figure 7C1–C4*, argue that GC B cells with centralized antigen (about one-third of cells) are the ones that make actomyosin arcs (again, about one-third of cells). We conclude, therefore, that GC B cells can make actomyosin arcs and that they likely use this structure to centralize antigen, although the degree to which they do this is considerably less than for naïve B cells. We note, however, that our conclusions regarding GC B cells require additional supporting data that include testing the ICAM-1 dependence of actomyosin arc formation and quantitating the contributions that this contractile structure makes to GC B cell traction forces, signaling, and antigen centralization.

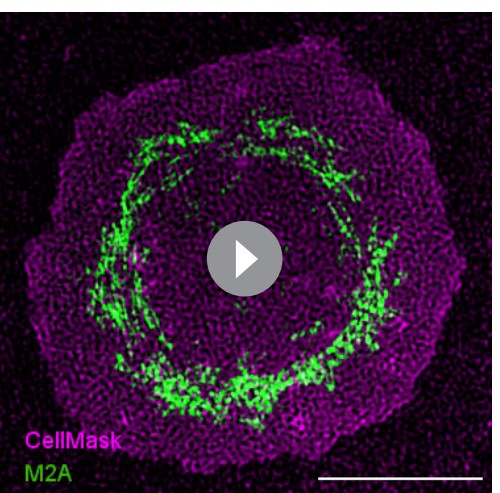

**Video 11.** A representative primary GC B cell isolated from the GFP-M2A knockin mouse that was stained with CellMask Deep Red (magenta) to label its plasma membrane, activated on glass coated with anti-IgM and ICAM-1, and imaged every 5 s for 300 s using TIRF-SIM. The first 11 frames show a still image of the magenta cell membrane. Played back at 10 fps. Scale bar: 5 μm.

https://elifesciences.org/articles/72805/figures#video11

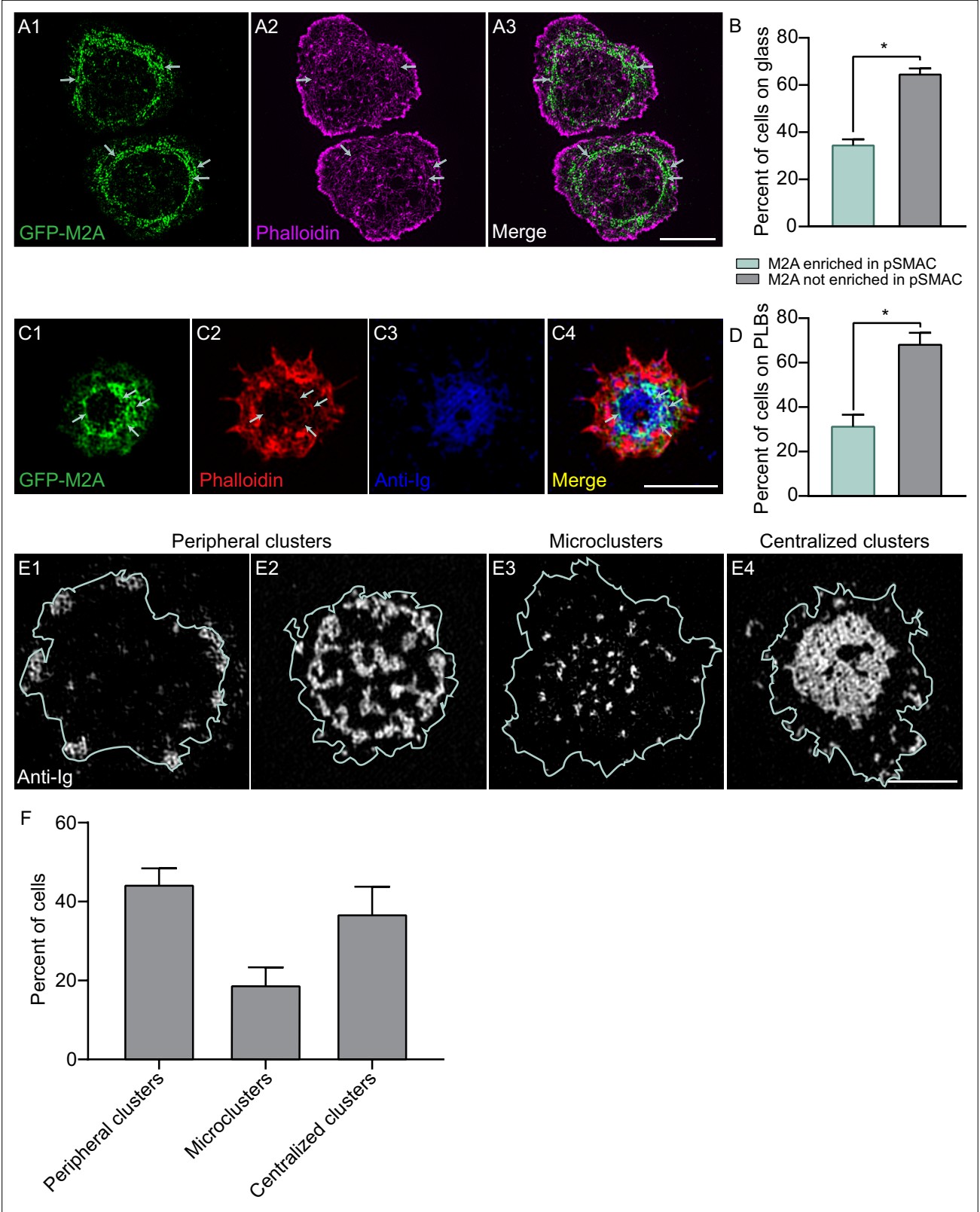

**Figure 7.** Germinal center B cells make actomyosin arcs. (**A1–A3**) Phalloidin-stained primary GC B cell from the M2A-GFP knockin mouse on anti-IgM/anti-IgG/ICAM-1-coated glass. White arrows mark the actomyosin arcs. (**B**) Percent of cells on glass that did or did not show M2A enrichment in the pSMAC (N = 140 cells from four experiments). (**C**) Phalloidin-stained primary GC B cell from the M2A-GFP knockin mouse 15 min after engagement with a PLB containing anti-IgM, anti-IgG, and ICAM-1. (**D**) Percent of cells on PLBs that did or did not show M2A enrichment in the pSMAC (N = 89 cells

*Figure 7 continued on next page*

from four experiments). (**E1–E4**) Representative images of the three types of anti-Ig distribution exhibited by GC B cells 15 min after engagement with a PLB containing anti-IgG and ICAM-1 (cell outlines are shown in blue). (**F**) Percent of GC cells displaying the three types of anti-Ig distribution shown in (**E1–E4**) (N = 157 cells from six experiments). All panels: TIRF-SIM images. Scale bars: 5 μm in (**A3**); 3 μm in (**C4, E4**).

The online version of this article includes the following figure supplement(s) for figure 7:

**Figure supplement 1.** Distribution of GFP-M2A in synapses formed by PLB-engaged germinal center B cells.

## Discussion

Integrin co-stimulation promotes B-cell activation and IS formation when antigen is limiting by promoting B-cell adhesion (*Carrasco et al., 2004*; *Carrasco and Batista, 2006b*). Here, we identified an actomyosin-dependent component of this integrin co-stimulatory effect. By combining super-resolution imaging with specific cytoskeletal perturbations, we showed that integrin ligation induces the formation of a pSMAC actomyosin arc network that comprises the major actin network at the primary B-cell IS. This network is created by the formin mDia1, organized into a concentric, contractile structure by the molecular motor M2A, and promotes synapse formation by mechanically sweeping antigen clusters centripetally into the cSMAC. Most importantly, we showed that integrin-dependent synapse formation under conditions of limiting antigen requires M2A as inhibiting its contractility significantly impairs antigen centralization. Consistently, myosin inhibition also diminishes the synaptic content of the key BCR signaling proteins P-CD79a and P-CD19 and disrupts their synaptic distribution. Finally, we showed that a significant fraction of GC B cells also make this contractile pSMAC actomyosin arc network. Together, our results argue that integrin co-stimulation promotes B-cell activation and synapse formation not only by enhancing B-cell adhesion (*Carrasco et al., 2004*), but also by eliciting the formation of a contractile actomyosin arc network that drives mechanical force-dependent IS formation. These findings invite a critical 'reset' for the way in which future B-cell studies should be approached by highlighting the need for integrin co-stimulation when examining the roles of actin and myosin during B-cell activation. This reset is especially important given that most in vitro studies of B-cell IS formation and activation have been performed under conditions of excess antigen, while antigen is rarely available in excess in vivo.

A central player in the link between integrin co-stimulation and the formation of the actomyosin arc network is almost certainly active RhoA. First, active RhoA would drive arc formation by simultaneously targeting, unfolding, and activating mDia1 at the plasma membrane (*Kühn and Geyer, 2014*;

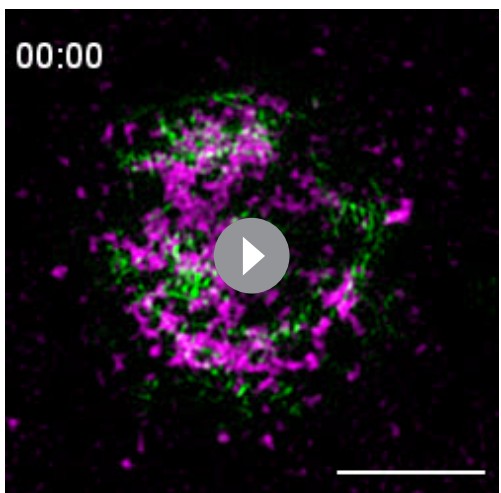

**Video 12.** A representative primary GC B cell isolated from the GFP-M2A knockin mouse that exhibited centralized antigen clusters on a PLB containing anti-Igs (magenta), imaged every 5 s for 300 s using TIRF-SIM and played back at 7 fps. Scale bar: 3 μm.
https://elifesciences.org/articles/72805/figures#video12

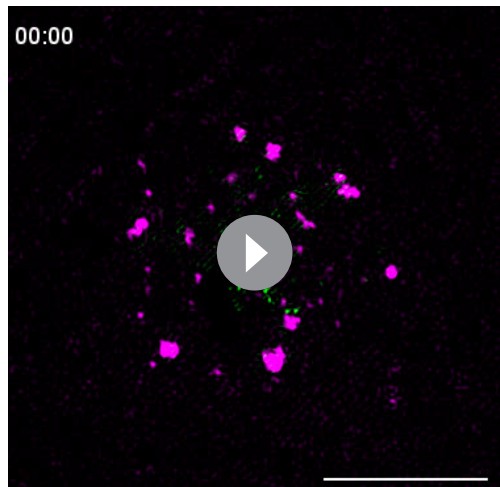

**Video 13.** A representative primary GC B cell isolated from the GFP-M2A knockin mouse that exhibited peripheral antigen clusters on a PLB containing anti-Igs (magenta), imaged every 5 s for 180 s using TIRF-SIM and played back at 7 fps. Scale bar: 5 μm.
https://elifesciences.org/articles/72805/figures#video13

*Rose et al., 2005*). Second, active RhoA would drive arc organization and contractility by activating the ROCK-dependent phosphorylation of the regulatory light chains on M2A (*Beach and Hammer, 2015*), thereby promoting the assembly of the M2A bipolar filaments that decorate, organize, and contract the arcs. Finally, it is likely that active RhoA would promote actomyosin arc formation by activating the ROCK-dependent phosphorylation of mDia1's autoinhibitory domain, thereby blocking its refolding and subsequent inactivation (*Nezami et al., 2006*; *Maiti et al., 2012*; *Staus et al., 2011*). Given all this, it seems very likely that integrin ligation promotes actomyosin arc formation at least in part by promoting the loading of RhoA with GTP. Consistent with this idea, adhesion signaling has been linked in a variety of systems to the activation of guanine nucleotide exchange factors (GEFs) for RhoA (e.g., p190RhoGEF, GEF H1) (*Guilluy et al., 2011*; *Lawson and Burridge, 2014*). Future work should seek, therefore, to clarify the outside-in signaling pathway in B cells that links integrin ligation to the activation of one or more GEFs for RhoA. Such efforts should also take into account parallel activation pathways, such as the PI3K-dependent activation of RhoA downstream of BCR signaling (*Saci and Carpenter, 2005*), the myosin-dependent activation of B-cell adhesion downstream of CXCR5 signaling (*Sáez de Guinoa et al., 2011*), and the diacylglycerol kinase-dependent regulation of adhesion and actomyosin force generation at the B-cell synapse (*Merino-Cortes, 2020*). Given our results here, the ability of the B-cell integrin VLA-4, which binds VCAM-1 on APCs, to promote IS formation under limiting antigen conditions (*Carrasco and Batista, 2006b*) may also involve an actomyosin-dependent mechanism. Indeed, actomyosin-dependent B-cell IS formation may be a mechanism harnessed by multiple co-stimulatory pathways to promote B-cell activation. Finally, future studies should also seek to clarify the extent to which integrin ligation promotes the formation of actomyosin arcs by driving their creation versus stabilizing them once created.

Consistent with our findings, a recent study by Bolger-Munro et al. reported that GFP-tagged M2A localizes to the medial portion of synapses formed by A20 B cells (*Bolger-Munro et al., 2019*). In their hands, however, BB treatment did not inhibit antigen centralization, arguing that synapse formation does not require M2A. The disparity between their results and ours as regards the functional significance of M2A may be due to numerous differences in experimental design, including the cell type used (primary B cells versus the A20 B cell line), the mode of antigen presentation (anti-IgM-containing PLBs versus transmembrane antigen expressed by APCs), and the density of antigen (known in PLBs versus unknown and variable on APCs). Our pSMAC actomyosin arcs may also be related to the myosin-rich regions that form in primary HEL-specific naïve B cells bound to acrylamide gels coated with HEL antigen (*Kumari et al., 2019*).

The contractile actomyosin structure identified here occupies the portion of the B-cell synapse defined by the presence of an integrin ring, that is, the pSMAC (*Harwood and Batista, 2011*; *Carrasco et al., 2004*; *Carrasco and Batista, 2006b*; *Dustin et al., 2010*). This co-localization should support a feed-forward relationship where integrin co-stimulation promotes the formation of the actomyosin arcs, and the contractile forces that these arcs then promote further integrin activation and robust adhesion. Indeed, the B-cell pSMAC can be viewed as roughly analogous to the lamellar region of mesenchymal cells, where integrins present within ECM-anchored focal adhesions are kept in their open, extended, high-affinity conformation by the forces that myosin-rich stress fibers exert on them (*Nordenfelt et al., 2016*; *Parsons et al., 2010*). By analogy, the contribution that the centering forces exerted by the actomyosin arcs make to integrin activation in the pSMAC may be enhanced in the context of an APC by the fact that the APC restricts ICAM-1 mobility (*Comrie et al., 2015b*). Of note, the activation of integrins by contractile actin arcs created by formins and myosin 2 is also seen in other cell types (*Tee et al., 2015*; *Tojkander et al., 2015*; *Burnette et al., 2014*).

Recent studies have presented evidence that GC B cells differ dramatically from naïve B cells with regard to the organization of actomyosin at their synapse (*Nowosad et al., 2016*; *Kwak et al., 2018*). We found, however, that about one-third of GC B cells exhibit robust actomyosin arcs in the medial, pSMAC portion of their synapse that are indistinguishable from those made by naïve B cells. Moreover, staining data, together with images of GFP-M2A distribution in synapses made by PLB-engaged GC B cells, suggest that, like naïve B cells, GC B cells can use this contractile structure to centralize antigen. Given that the selection of GC B cells with higher affinity BCRs likely depends to a significant extent on their ability to gather antigen in the context of strong competition for limiting antigen presented by follicular dendritic cells (*Heesters et al., 2016*; *Heesters et al., 2014*), we suggest that actomyosin arcs might contribute to this selection process by promoting antigen gathering.

The actomyosin arcs described here in B cells and the actomyosin arcs described previously in T cells (*Murugesan et al., 2016*) have a great deal in common as regards their formation, organization, and dynamics (*Wang and Hammer, 2020*; *Hammer et al., 2019*). Consistently, this contractile structure supports a number of synaptic processes that are shared by these two cell types, including antigen centralization, proximal signaling, and the formation of an adhesive ring in the medial portion of the IS. A major question, then, is how these two immune cell types harness the force generated by this shared contractile structure to perform their unique functions, that is, target cell killing by the T cell and antibody creation by the B cell. Stated another way, how does the T cell use the force generated by this contractile structure to support the effectiveness of an exocytic event (lytic granule secretion), while the B cell uses the force to support the effectiveness of an endocytic event (antigen extraction and uptake). With regard to T cells, a seminal study by *Basu et al., 2016* showed that actomyosin-dependent forces placed on the target cell membrane by the T cell increase the efficiency of target cell killing by straining the target cell membrane in such a way as to increase the pore-forming activity of perforin. One clear goal, therefore, is to determine if the T cell's actomyosin arcs are responsible for creating this strain. Imaging the actomyosin arcs during the process of target cell killing, and blocking the force they generate just prior to lytic granule secretion, should reveal their contribution to this essential effector function.

The idea that B cells would use the actomyosin arcs identified here to support the extraction and endocytic uptake of membrane-bound antigens stems from the seminal work of Tolar and colleagues, who showed that M2A plays an important role in antigen extraction (*Nowosad et al., 2016*; *Natkanski et al., 2013*; *Hoogeboom, 2018*). These authors also presented evidence that M2A-dependent pulling forces select for BCRs with higher affinity for antigen as such interactions survive the myosin-dependent strain placed on them, resulting in antigen extraction (*Nowosad et al., 2016*; *Natkanski et al., 2013*; for review, see *Wang and Hammer, 2020* and *Spillane and Tolar, 2018*). That said, a recent, imaging-based effort to define the mechanism by which antigen is extracted did not provide clear insight into how M2A contributes to this process. Specifically, *Roper et al., 2019* reported that the synapses of naïve B cells bound to antigen-bearing plasma membrane sheets (PMSs) are composed of a dynamic mixture of actin foci generated by the Arp2/3 complex and disorganized linear filaments/fibers generated by a formin. While static images showed little co-localization between the actin foci and antigen clusters, dynamic imaging suggested that the foci promote antigen extraction (although formin activity was also required). Based on these and other observations, Roper et al. concluded that naïve B cells use a foci-filament network to drive force-dependent antigen extraction (*Roper et al., 2019*). How M2A contributes to this force was unclear, however, as M2A (visualized using an antibody to the phosphorylated form of M2A's RLC) did not co-localize with either actin structure (*Roper et al., 2019*). Moreover, neither actin structure was affected by BB treatment. These two findings are notably at odds with our findings that M2A (visualized by endogenous tagging of the M2A heavy chain) co-localizes dramatically with actin arcs, and that BB treatment profoundly disrupts the organization of the pSMAC actin arc network. Regarding this discrepancy, we note that the images of synaptic actin presented by Roper et al. look similar to our images of naïve B cells stimulated with anti-IgM alone, where the synapse was also composed of a disorganized and dynamic mixture of actin foci and short-actin filaments/fibers. The fact that the PMSs used by Roper et al. did not contain integrin ligands may explain, therefore, why they did not see a more organized synapse containing actomyosin arcs. In the same vein, two other recent studies examined antigen extraction using substrates that lacked integrin ligands (PLBs and PMSs in *Kwak et al., 2018* and acrylamide gels in *Kumari et al., 2019*). Given our results, we suggest that future efforts to define the mechanism by which M2A promotes antigen extraction should follow the myosin as the B cell extracts antigen from an APC, where the B cell's integrins will be engaged, and where the antigen can be presented in a physiologically relevant way (e.g., opsonized and bound to an Fc or complement receptor). Such efforts will hopefully reveal how the B cell harnesses the forces generated by the actomyosin arcs identified here to drive antigen extraction and uptake.

## Materials and methods

### Mice and cell culture

Primary B cells were isolated from the spleens of 6–12-week-old C57BL/6 mice (Jackson Laboratories #002595) and GFP-M2A (*Myh9*) KI mice (gift of R. Adelstein, NHLBI/NIH) of either sex using negative selection B cell isolation (StemCell Technologies). Euthanasia was performed in accordance with protocols approved by the National Human Genome Research Institute Animal Use and Care Committee at the National Institutes of Health. The A20 murine IgG⁺ B cell line was purchased from ATCC (ATCC TIB-208), verified by responsiveness to anti-IgG stimulation, and confirmed to be free of mycoplasma. B cells were cultured in complete medium (RPMI-1640, 10% heat-inactivated fetal calf serum [FCS], 2 mM L-glutamine, 1 mM sodium pyruvate, 50 µM 2-mercaptoethanol, and 1X Antibiotic-Antimycotic) at 37°C with 5% $CO_2$. Primary B-cell complete media also contains 5 ng/ml of BAFF (R&D Systems).

### Plasmids and reagents

GFP- and tdTomato-tagged F-Tractin were gifts from Michael Schell (Uniformed Services University, Maryland). Alexa Fluor-conjugated phalloidins were purchased from Thermo Fisher. Anti-mDia1 antibody was purchased from Thermo Fisher (PA5-27607). HRP-conjugated mouse anti-β-actin antibody was purchased from Santa Cruz (SC-47778 HRP). Rabbit anti-CD79a (#3351), anti-PCD79a (#5173), anti-CD19 (#3574), and anti-PCD19 (#3571) were purchased from Cell Signaling Technologies. Anti-M2A was purchased from MilliporeSigma (#M8064). CK-666 and SMIFH2 were purchased from MilliporeSigma and used at final concentrations of 100 µM and 25 µM, respectively. pnBB and paBB were purchased from Cayman Chemicals and used at a final concentration of 25 µM. DMSO vehicle control was purchased from MilliporeSigma. CellMask Deep Red Plasma Membrane Stain was purchased from Thermo Fisher. Alexa Fluor 488- (#111-545-003), 594- (#111-585-003), and 647- (#111-605-003) conjugated goat, anti-rabbit secondary antibodies were purchased from Jackson ImmunoResearch. Goat anti-mouse IgG Fcγ fragment-specific antibody (#115-005-008) and goat anti-mouse IgM, µ-chain-specific antibodies (#115-005-020) were purchased from Jackson ImmunoResearch. Anti-rabbit-HRP (#32260) was purchased from Thermo Fisher.

### GC B-cell generation and sorting

GC B cells were generated and sorted using a previously described protocol (*Hwang et al., 2015*). Briefly, 6–12-week-old M2A-GFP KI mice were immunized with sheep's red blood cells. After 8–10 days, total B cells from the spleens and lymph nodes were isolated using the Negative Selection B cell isolation kit (StemCell Technologies) according to the manufacturer's instructions. Dead cells were stained using Zombie Yellow viability stain (BioLegend) and Fc receptors were blocked with the mouse TruStain FcX antibody (#156604). Cells were immunostained with anti-mouse CD38 (#102719), B220 (#103235), and GL-7 (#144617) purchased from BioLegend. GC B cells were sorted on a BD Aria III FACs sorter (Beckton Dickinson) for GFP+, Zombie Yellow⁻, B220⁺, CD38^low and GL-7⁺ cells, and were used immediately.

### B-cell transfection

A20 B cells and primary B cells were transfected as previously described (*Wang et al., 2017*). Briefly, ex vivo primary B cells were first cultured for 12 hr in complete media supplemented with 5 ng/ml BAFF (R&D Systems) and 2.5 µg/ml *Escherichia coli* O111:B4 LPS (MilliporeSigma) (LPS was included to promote cell survival during nucleofection). $2 \times 10^6$ B cells were then nucleofected with 2 µg of plasmid DNA using Nucleofector Kit V (Lonza) and rested for at least 16–24 hr using complete media containing 5 ng/ml BAFF and lacking LPS. We refer to both rested, transfected cells and ex vivo nonmanipulated cells as naïve B cells because neither had been activated by antigen.

### CRISPR

Mouse GFP-M2A and Scarleti-M2A template plasmids were gifts from Jordan Beach (Loyola University, Chicago). Mouse M2A sgRNAs were synthesized by Synthego and used according to the manufacturer's instructions. Briefly, sgRNAs were mixed with Cas9 (IDT) to form ribonucleoproteins and then added together with 0.5 µg of template plasmid to $2 \times 10^6$ cells suspended in the solution

for Nucleofector Kit V. Following nucleofection, the cells were cultured in complete media for 24 hr before fluorescence-activated cell sorting (FACS) for GFP or Scarleti expression using the Aria III (Becton Dickinson).

## miRNA-mediated knockdown of mDia1

miRNAs targeting the 3′ UTR of mouse mDia1 were designed as previously described (*Alexander and Hammer, 2016*) using BLOCK-iT RNAi Designer (Thermo Fisher), synthesized (Gene Universal), and fused to the C-terminus of mNeonGreen-F-Tractin using In-Fusion cloning (Takara). As a control, a version of this plasmid containing an miRNA sequence that has been verified as nontargeting in mouse (*Cai et al., 2006*) was used. A20 B cells were transfected with 2 µg of F-Tractin-mNeonGreen vector control, F-Tractin-mNeonGreen-mDia1-miRNAs, F-Tractin-mNeonGreen-nontargeting miRNA, or a combination of F-Tractin-mNeonGreen and 300 nM of mirVana-negative control and cultured in complete media for 16 hr. Cells were then lysed and immunoblotted using an antibody to mDia1 (1:250) and an HRP-conjugated antibody to β-actin (1:5000) to confirm knockdown. Cells that had received the miRNA were identified based on the expression of F-Tractin-mNeonGreen and then quantified after staining with phalloidin. F-Tractin-mNeonGreen-positive cells were also used in a cell spreading assay as described below.

## Cell spreading on functionalized glass

Eight-well Labtek chambers (Nunc) were coated with 15 µg/ml of anti-IgM and/or anti-IgG with or without 0.5 µg/ml of mouse histidine-tagged ICAM-1 (Sino Biological) for 1 hr at room temperature. B cells were resuspended in modified HEPES-buffered saline (mHBS) (*Wang et al., 2017*) and adhered to functionalized glass for 15 min at 37°C before live imaging or fixing with 4% paraformaldehyde for staining (see Materials and methods). Where inhibitors were used, cells were pretreated for 30 min with 100 µM CK-666, 25 µM SMIFH2, 25 µM pnBB or paBB, or dH₂O/DMSO vehicle control in mHBS at 37°C. Cells were then added to functionalized Labtek chambers in mHBS containing the same concentrations of inhibitors or vehicle control as the pretreatment.

## Supported planar lipid bilayers

Liposomes were prepared as described previously (*Yi et al., 2012*; *Murugesan et al., 2016*; *Hong et al., 2017*). Briefly, 0.4 mM 1,2-dioleoyl-*sn*-glycero-3-phosphocholine, biotin–CAP-PE, 1,2-dioleoyl-*sn*-glycero-3-[(*N*-(5-amino-1-carboxypentyl)iminodiacetic acid)succinyl] (DGS)–NTA and 1,2-dioleoyl-sn-glycero-3-phosphocholine (Avanti Polar Lipids, Inc) were mixed at 1:3:96 molar % ratio. Lipids were dried under a stream of argon and then desiccated in a vacuum chamber. Unilamellar liposomes were generated from lyophilized lipids hydrated in Tris-buffered saline via extrusion through a 50 nm pore membrane using a mini-extruder kit (Avanti Polar Lipids, Inc). PLBs were assembled in Sticky-Slide VI$^{0.4}$ Luer closed chambers (Ibidi) as previously described (*Comrie et al., 2015a*). 25 × 75 mm glass coverslips (Ibidi) were cleaned using Piranha solution (1:3 ratio of sulfuric acid and 30% hydrogen peroxide). After depositing liposomes onto the flow channels, the channels were washed with HBS buffer containing 1% BSA. A solution containing mono-biotinylated, Alexa Fluor 647-labeled anti-IgM antibody (0.15 µg/ml [300 molecules/µm²] for the limiting antigen condition and 2 µg/ml [4000 molecules/µm²] for the high antigen condition) and streptavidin (Sigma-Aldrich) were added to the flow chambers with or without 0.5 µg/ml unlabeled histidine-tagged ICAM-1. Anti-IgM antibody (µ-chain specific) was monobiotinylated and labeled with Alexa Fluor 647 (Thermo Fisher) as described previously (*Carrasco et al., 2004*). The uniformity and lateral mobility of PLBs were assessed using FRAP as described previously (*Yi et al., 2012*). Photobleached circles with a diameter of 4 µm typically recovered within 60 s. B cells were resuspended in modified HEPES-buffered saline and allowed to engage PLBs at 37°C and imaged immediately, or fixed with 4% paraformaldehyde after 5 and 10 min for immunostaining.

## Traction force microscopy

Polyacrylamide gels (PA, 0.23 kPa shear modulus, 40 µm thickness) were prepared on glass coverslips with embedded 40 nm fluorescent beads (TransFluoSpheres [633/720], Thermo Scientific), as described previously (*Jaumouillé et al., 2019*). B cells were resuspended in mHBS with 2% FCS and added to PA gels. Images of B cells that had engaged PA gels for 20 min were captured. A no-stress

reference image of the PA gels with beads was captured after lifting cells from the PA gel by adding 1% sodium dodecyl sulfate in 1× PBS to the imaging chamber at a final concentration of 0.04%. Particle image velocimetry was used to calculate bead displacements relative to the reference position, and the corresponding contractile energy was quantified using ImageJ plugins as previously described (*Jaumouillé et al., 2019*; *Martiel et al., 2015*). Traction forces were reported as the mean magnitude of traction stress within the cell relative to the cell surface area.

## Immunostaining

Fixed cells were permeabilized with 0.2% Triton-X-100 and blocked for 30 min at room temperature using PBS containing 2% BSA. Cells were incubated with primary antibodies (1:200) overnight at 4°C and then secondary antibodies (1:250) with Alexa Fluor-conjugated phalloidins for 1 hr at room temperature. Antibodies and phalloidins were diluted in blocking buffer. All washes were performed with 1× PBS.

## Microscopy

All live-cell imaging was performed at 37°C in mHBS supplemented with 2% FCS. TIRF-SIM and 3D-SIM imaging were performed on a GE DeltaVision OMX SR microscope (Cytiva) equipped with a 60 × 1.42 NA oil objective (Olympus). For 3D-SIM, z-stacks were acquired at 0.125 µm increments. Raw data were reconstructed using Softworx software (Cytiva) with a Wiener filter constant of 0.002–0.003. Airyscan imaging was performed using an LSM 880 Zeiss confocal microscope equipped with Airyscan and using a Plan-Apochromat 63 × 1.4 NA oil objective. Airyscan image reconstruction was performed using Zeiss ZEN imaging software. TFM was imaged using a Nikon Eclipse Ti2 microscope equipped with a 60 × 1.2 NA water objective. Linear adjustments to images were made using ImageJ 1.53 (NIH).

## Image analyses

All image analyses were performed using ImageJ (NIH). To draw ROIs for measurements of the fraction of total IS footprint occupied by each SMAC, the content of F-actin in each SMAC, and the anisotropy of actin filaments within the pSMAC, we relied on the distinctive appearance of actin in each SMAC. This was straightforward in TIRF-SIM images of B cells stimulated with both anti-IgM and ICAM-1, where the thin outer dSMAC was comprised of moderately bright pixels with not much fluctuation in intensity, the medial pSMAC was comprised of bright actin arcs with intervening dim signals, and the central cSMAC was comprised mostly of dim signals. For B cells engaged with anti-IgM alone, the thin outer dSMAC was still readily identifiable, the central cSMAC was identifiable in lower mag images as a central circle with less signal than the area between it and the dSMAC, and the medial pSMAC corresponded to the area between the dSMAC and the cSMAC. Fluorescence intensities within the SMAC regions were quantified using ROIs and reported as the total background-corrected fluorescence within the ROI, which was quantified as described (*Burgess et al., 2010*) using the following equation: Integrated density – [(area of ROI) × (mean background fluorescence per unit area)], where the integrated density is equal to [(area of ROI) × (mean fluorescence per unit area within the ROI)]. Mean background fluorescence was determined using the same ROI size at three separate positions less than 3 µm away from the cell. The myosin fluorescence intensity in 3D-SIM images was quantified using a maximum projection image of the image stacks where the cell ROI was determined based on the F-actin threshold and the background-corrected myosin fluorescence within the cell ROI was reported. The FibrilTool plugin for ImageJ was used to measure actin arc morphology based on the intensity gradients between pixels as described previously (*Murugesan et al., 2016*; *Boudaoud et al., 2014*). Briefly, the pSMAC regions in TIRF-SIM images were divided into 10–12 trapezoid-shaped ROIs of similar size to measure the anisotropy of arcs in the radially symmetric pSMAC. The values obtained range from 0, when the orientation of the structures is random, to 1, when the structures show higher orientation in the same direction. The velocity of centripetal actin flow was assessed by assembling kymographs from TIRF-SIM videos using the Kymograph Builder plugin from ImageJ, as previously described (*Murugesan et al., 2016*). Briefly, the dSMAC and pSMAC regions were identified by the relatively abrupt slope change for F-actin flow, and slope angles were used to quantify the rates of actin movement. The size of each antigen cluster and their relative distance from the cSMAC center were quantified using an ImageJ macro. First, the perimeter of the synaptic interface

was determined based on thresholds for F-actin, and an ROI that encompassed the interface area was drawn (the synaptic ROI). The coordinates for each pixel contained in the ROI were determined and the linear distance of each pixel from the center of mass of the total synaptic antigen (defined as cSMAC center) was determined. The longest distance was defined as the furthest distance to travel from the outermost edge of the cell. A binary image of the antigen channel combined with the ImageJ watershed algorithm was used to segment individual antigen clusters within the synaptic ROI. The area of each antigen cluster was quantified using the Analyze Particles function in ImageJ. The relative distance of each antigen cluster was reported as the distance between the center of mass of the antigen cluster and the cSMAC center after normalizing to the furthest distance from the cell edge to the cSMAC center. To quantify the antigen fluorescence in the cSMAC, a circular ROI corresponding to 20% of the total synaptic area (based on the average area of the cSMAC at the synaptic interface) was drawn such that the center of the circle lies at the same coordinates as the center of mass of the total antigen signal. Antigen fluorescence within this circle was quantified and presented as a percent of the total synaptic antigen fluorescence. The fluorescence intensities of the signaling molecules CD79a, P-CD79a, CD19, and P-CD19 were all reported as the total fluorescence intensity within the synaptic ROI. All fluorescence intensities were corrected for background as described above. Fluorescence intensity profiles were obtained by drawing a 10 µm line across the center of the synaptic interface and using the ImageJ function 'Plot Profiles' to obtain fluorescence intensity values across the line. The intensity profiles of several cells were combined, and the average fluorescence intensity ± standard deviation was reported. The speeds of antigen cluster movement were quantified using the ImageJ plugin TrackMate as previously described (*Tinevez et al., 2017*) where a combination of automated and manual tracking was performed. Prior to quantification, the perimeter of the cell was identified by oversaturating the signal for GFP-M2A, and the anti-IgM fluorescence signal outside of the cell was removed so that only antigen clusters formed by that cell were quantified. Antigen clusters were determined using a blob diameter of 0.2 µm², and tracks were obtained using a threshold of 2000 units with sub-pixel localization. Mean antigen cluster movement speeds were reported as distance traveled over time. Kymographs of moving antigen clusters were created using the ImageJ plugin Kymograph Builder.

## Statistical analyses

All statistical analyses were performed using Prism 9 (GraphPad). Statistical comparisons of dot plots were performed using unpaired, two-tailed *t*-tests, and data are represented as mean ± standard deviation. Statistical comparisons of bar charts were performed using paired, two-tailed *t*-tests, and data are represented as mean ± standard error of the mean, unless otherwise stated. The following annotations are used to indicate significance: *p<0.05, **p<0.01, ***p<0.001, and ****p<0.0001.

## Acknowledgements

This work was supported by the Intramural Research Program of the National Heart, Lung, and Blood Institute (NHLBI) (1ZIAHL006121-04 to JAH). The authors thank the NHLBI Flow Cytometry Core, Dr. Xuefei Ma for providing M2A-GFP mice, Dr. Il-Young Hwang for immunizing mice to prepare GC B cells, and Dr. Christopher J Alexander for advice on miRNA design.

## Additional information

### Funding

| Funder | Grant reference number | Author |
| --- | --- | --- |
| National Heart, Lung, and Blood Institute | 1ZIAHL006121-04 | John A Hammer |

The funders had no role in study design, data collection and interpretation, or the decision to submit the work for publication.

## Author contributions
Jia C Wang, Conceptualization, Data curation, Formal analysis, Investigation, Methodology, Project administration, Software, Validation, Visualization, Writing – original draft, Writing – review and editing; Yang-In Yim, Resources; Xufeng Wu, Investigation; Valentin Jaumouille, Data curation, Formal analysis, Investigation; Andrew Cameron, Data curation, Resources; Clare M Waterman, Methodology, Resources; John H Kehrl, Conceptualization, Funding acquisition, Methodology, Project administration, Resources, Supervision, Writing – original draft, Writing – review and editing; John A Hammer, Conceptualization, Data curation, Funding acquisition, Project administration, Resources, Supervision, Writing – original draft, Writing – review and editing

## Author ORCIDs
Jia C Wang http://orcid.org/0000-0002-8666-4662
John H Kehrl http://orcid.org/0000-0002-6526-159X
John A Hammer http://orcid.org/0000-0002-2496-5179

## Ethics
This study was performed in strict accordance with the recommendations and protocols approved by the National Human Genome Research Institute Animal Use and Care Committee at the National Institutes of Health. All animals were handled according to approved institutional animal care and use committee (IACUC) protocols (#H-0337) of the National Institutes of Health.

## Decision letter and Author response
Decision letter https://doi.org/10.7554/eLife.72805.sa1
Author response https://doi.org/10.7554/eLife.72805.sa2

---

## Additional files

### Supplementary files
• Transparent reporting form

### Data availability
All data generated or analyzed during this study are included in the manuscript and supporting files. Numerical data have been provided as source data and are available from the Dryad database.

The following dataset was generated:

| Author(s) | Year | Dataset title | Dataset URL | Database and Identifier |
|-----------|------|---------------|-------------|-------------------------|
| Wang JC | 2022 | A B cell actomyosin arc network couples integrin co-stimulation to mechanical force-dependent immune synapse formation | https://doi.org/10.5061/dryad.9kd51c5km | Dryad Digital Repository, 10.5061/dryad.9kd51c5km |

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

# Appendix 1

## Appendix 1—key resources table

| Reagent type (species) or resource | Designation | Source or reference | Identifiers | Additional information |
|---|---|---|---|---|
| Strain, strain background (*Mus musculus*) | C57BL/6 | Jackson Laboratories | Cat# 002595; RRID:MGI:5656552 | |
| Strain, strain background (*M. musculus*) | M2A-GFP KI | Robert Adelstein, NHLBI/ NIH | | |
| Cell line (*M. musculus*) | A20 | American Type Culture Collection | ATCC TIB-208; RRID:CVCL_1940 | IgG[+] B cell line |
| Recombinant DNA reagent | GFP-F-Tractin | Michael Schell (Uniformed Services University, MD) | | |
| Recombinant DNA reagent | tdTomato-F-Tractin | Michael Schell (Uniformed Services University, MD) | | |
| Chemical compound, drug | Alexa Fluor 488 Phalloidin | Thermo Fisher Scientific | Cat# A12379 | Fluorescence labeling (1:500) |
| Chemical compound, drug | Alexa Fluor 568 Phalloidin | Thermo Fisher Scientific | Cat# A12380 | Fluorescence labeling (1:500) |
| Chemical compound, drug | CellMask Deep Red Plasma Membrane Stain | Thermo Fisher Scientific | Cat# C10046 | Cell labeling (1:10000) |
| Chemical compound, drug | DMSO | MilliporeSigma | Cat# D4540 | |
| Chemical compound, drug | SMIFH2 | MilliporeSigma | Cat# 344092 | 25 µM |
| Chemical compound, drug | CK-666 | MilliporeSigma | Cat# 182515 | 100 µM |
| Chemical compound, drug | (S)-nitro-blebbistatin (pnBB) | Cayman Chemicals | Cat# 24171 | 25 µM |
| Chemical compound, drug | Para-amino blebbistatin | Cayman Chemicals | Cat# 22699 | 25 µM |
| Chemical compound, drug | Zombie Yellow viability stain | BioLegend | Cat# 423103 | Dead cell staining (1:300) |
| Chemical compound, drug | 18:1 Biotinyl Cap PE IN CHLOROFORM 1,2-dioleoyl-sn-glycero-3-phosphoethanolamine-N-(cap biotinyl) | Avanti Polar Lipids | Cat# 870273C | |
| Chemical compound, drug | 18:1 DGS-NTA(Ni) in Chloroform 1,2-dioleoyl-sn-glycero-3-[(N-(5-amino-1-carboxypentyl) iminodiacetic acid)succinyl] (nickel salt) | Avanti Polar Lipids | Cat# 790404C | |
| Chemical compound, drug | 18:1 (9-Cis) PC (DOPC) in CHLOROFORM 1,2-dioleoyl-sn-glycero-3-phosphocholine | Avanti Polar Lipids | Cat# 850375C | |
| Chemical compound, drug | TransFluoSpheres (633/720) | Thermo Fisher Scientific | Cat# T8870 | |
| Other | *Escherichia coli* O111:B4 LPS | MilliporeSigma | Cat# L2630 | Cell culture 2.5 µg/ml |
| Biological sample (*Ovis aries*) | Sheep's red blood cells | Innovative Research Novi | Cat# ISHRBC100P15ML | Injection $2 \times 10^8$ cells |
| Antibody | Alexa Fluor 488- conjugated goat, anti-rabbit, polyclonal | Jackson Immuno Research | Cat# 111-545-003; RRID:AB_2338046 | Immunofluorescence (1:500) |

*Appendix 1 Continued on next page*

*Appendix 1 Continued*

| Reagent type (species) or resource | Designation | Source or reference | Identifiers | Additional information |
|---|---|---|---|---|
| Antibody | Alexa Fluor 594 conjugated goat, anti-rabbit, polyclonal | Jackson Immuno Research | Cat# 111-585-003; RRID:AB_2338059 | Immunofluorescence (1:500) |
| Antibody | Alexa Fluor 647- conjugated goat, anti-rabbit, polyclonal | Jackson Immuno Research | Cat# 111-605-003; RRID:AB_2338072 | Immunofluorescence (1:500) |
| Antibody | Goat anti-mouse IgG, Fcγ fragment-specific, polyclonal | Jackson Immuno Research | Cat# 115-005-008; RRID:AB_2338449 | Coverslip coating 2.5 µg/cm$^2$ |
| Antibody | Goat anti-mouse IgM, µ-chain-specific, polyclonal | Jackson Immuno Research | Cat# 115-005-020; RRID:AB_2338450 | Coverslip coating 2.5 µg/cm$^2$ |
| Antibody | Goat anti-rabbit IgG (H + L) Poly-HRP, polyclonal | Thermo Fisher | Cat# 32260; RRID:AB_1965959 | Western blot (1:3000) |
| Antibody | Rabbit anti-DIAPH1, polyclonal | Thermo Fisher | Cat# PA5-27607; RRID:AB_2545083 | Western blot (1:250) |
| Antibody | β-actin antibody (C4), mouse monoclonal | Santa Cruz | Cat# SC-47778 HRP; RRID:AB_2714189 | Western blot (1:5000) |
| Antibody | Rabbit anti-CD79a, polyclonal | Cell Signaling Technologies | Cat# 3351; RRID:AB_2075745 | Immunofluorescence (1:250) |
| Antibody | Rabbit anti-phospho-CD79a, polyclonal | Cell Signaling Technologies | Cat# 5173; RRID:AB_10694763 | Immunofluorescence (1:250) |
| Antibody | Rabbit anti-CD19, polyclonal | Cell Signaling Technologies | Cat# 3574; RRID:AB_2275523 | Immunofluorescence (1:250) |
| Antibody | Rabbit anti-phospho-CD19, polyclonal | Cell Signaling Technologies | Cat# 3571; RRID:AB_2072836 | Immunofluorescence (1:250) |
| Antibody | Rabbit anti-M2A, polyclonal | MilliporeSigma | Cat# M8064; RRID:AB_260673 | Immunofluorescence (1:200) |
| Antibody | TruStain FcX PLUS (anti-mouse CD16/32 antibody, rat monoclonal | BioLegend | Cat# 156604; RRID:AB_2783138 | FcR block (0.25 µg/10$^6$ cells) |
| Antibody | Pacific Blue anti-mouse CD38, rat monoclonal | BioLegend | Cat# 102719; RRID:AB_10613289 | FACS (1:100) |
| Antibody | PerCP/Cyanine5.5 anti-mouse/human CD45R/B220, rat monoclonal | BioLegend | Cat# 103235; RRID:AB_893356 | FACS (1:100) |
| Antibody | APC anti-MU/HU GL7 antigen, rat monoclonal | BioLegend | Cat# 144617; RRID:AB_2800674 | FACS (1:200) |
| Recombinant DNA reagent | mNeonGreen-F-Tractin | This paper | | See Materials and methods |
| Sequence-based reagent | Non-targeting miRNA | This paper | miRNA | ACCTAAGGTTA AGTCGCCCTCG; see also Materials and methods |
| Sequence-based reagent | mDia1 miRNA #1 | This paper | miRNA | CAGCATGGCT AAATGGTCA; see also Materials and methods |
| Sequence-based reagent | mDia1 miRNA #2 | This paper | miRNA | GGGTCCGTTT GCTGCCTTA; see also Materials and methods |
| Sequence-based reagent | mDia1 miRNA #3 | This paper | miRNA | GGGTAGCAAT GCTGTGTTT; see also Materials and methods |
| Sequence-based reagent | mirVana miRNA Mimic, Negative Control #1 | Thermo Fisher Scientific | Cat# 4464058 | |
| Sequence-based reagent | MYH9 sgRNA | Synthego | sgRNA | AAACUUCAUCA AUAACCCGC |
| Recombinant DNA reagent | Mouse GFP-M2A | Jordan Beach (Loyola University, Chicago) | | CRISPR GFP-M2A template |
| Recombinant DNA reagent | Mouse Scarleti-M2A | Jordan Beach (Loyola University, Chicago) | | mScarleti-CRISPR M2A template |

*Appendix 1 Continued on next page*

*Appendix 1 Continued*

| Reagent type (species) or resource | Designation | Source or reference | Identifiers | Additional information |
|---|---|---|---|---|
| Peptide, recombinant protein | Alt-R S.p. HiFi Cas9 Nuclease V3 | IDT | Cat# 1081060 | |
| Peptide, recombinant protein | BAFF | R&D Systems | Cat# 8876-BF-010 | Cell culture 5 ng/ml |
| peptide, recombinant protein | Streptavidin | MilliporeSigma | Cat# 189730 | |
| peptide, recombinant protein | Mouse histidine-tagged ICAM-1 | Sino Biological | Cat# 50440-M08H | |
| Commercial assay or kit | Nucleofector Kit V | Lonza | Cat# VCA-1003 | |
| Commercial assay or kit | In-Fusion HD Cloning | Takara | Cat# 638911 | |
| Commercial assay or kit | Mini-extruder kit | Avanti Polar Lipids | Cat# 610000 | |
| Commercial assay or kit | Sticky-Slide VI$^{0.4}$ Luer closed chambers | Ibidi | Cat# 80608 | |
| Commercial assay or kit | Alexa Fluor 647 Antibody Labeling Kit | Thermo Fisher Scientific | Cat# A20186 | |
| Commercial assay or kit | EZ-Link Micro Sulfo-NHS-Biotinylation Kit | Thermo Fisher Scientific | Cat# 21925 | |
| Software, algorithm | ImageJ | NIH | | |
| Software, algorithm | Fiji | *Schindelin et al., 2012* | RRID:SCR_002285 | https://imagej.net/Fiji |
| Software, algorithm | Softworx | Applied Precision Ltd.; GE Healthcare Life Sciences | RRID:SCR_019157 | |
| Software, algorithm | ZEN | Zeiss | RRID:SCR_018163 | |
| Software, algorithm | FibrilTool | *Boudaoud et al., 2014* | RRID:SCR_016773 | |
| Software, algorithm | BLOCK-iT RNAi Designer | Thermo Fisher Scientific | RRID:SCR_002794 | https://rnaidesigner.thermofisher.com/rnaiexpress/ |
| Software, algorithm | Prism | GraphPad | RRID:SCR_002798 | |
| Software, algorithm | Traction Force plugin | *Martiel et al., 2015* | | |

