## [Editor Report]

This study has used striking live-cell super-resolution microscopy methods to demonstrate the direct relationship between integrin dependent F-actin/myosin arcs and transport of surface immunoglobulin-antigen clusters in B-cell synapses. The function of the F-actin arcs in signaling at limiting antigen levels where integrins are important is also demonstrated. In addition, this study has shown that both follicular and germinal center B cells utilize the F-actin arcs, suggesting that this machinery can operate in both initiation and affinity maturation phases of antibody-based adaptive immunity. The work will be of interests to immunologists, cell biologists, and biophysicists, and the data sets should be of future use in modeling the process.

---

## [Decision Letter]

**Decision letter after peer review:**

Thank you for submitting your article "A B cell actomyosin arc network couples integrin co-stimulation to mechanical force-dependent immune synapse formation" for consideration by *eLife*. Your article has been reviewed by 3 peer reviewers, and the evaluation has been overseen by a Reviewing Editor and Anna Akhmanova as the Senior Editor. The following individual involved in review of your submission has agreed to reveal their identity: Pieta Mattila (Reviewer #2).

Essential revisions:

Your paper is of interest to immunologists studying mechanisms of lymphocyte activation and scientists in the broader field of cell mechanics. The work provides new insight into the cooperation among receptors, the actin cytoskeleton, and myosin motors that is required for the formation of a B cell immune synapse. The data support the key claims of the manuscript.

There are a few instances where more complete analysis and presentation of results would strengthen the paper.

1) The requirement for LFA-1:ICAM-1 ligation in the formation of the actomyosin arcs is not clear. The authors observe that ~30% of B cells form actomyosin arcs with anti-IgM stimulation only (Figure 1). Does LFA-1:ICAM-1 ligation simply stabilise the arcs and therefore make their appearance more likely, or does it promote the formation of a distinct actomyosin network with unique functions? The images and videos selected to represent cells stimulated with anti-IgM only (Figure 1; Videos 1A and 1B) seem form a highly branched actin network throughout the synapse, but it would be informative to see cells having the actomyosin arcs for comparison. Since B cells stimulated with anti- IgM alone are capable of signalling and centralising antigen, it would be interesting to know whether and how these two populations (with and without arcs) differ.

2) The observation that some GC B cells centralise antigen is very interesting, but there are a few aspects of this investigation that should be expanded upon. The authors show that with LFA-1:ICAM-1 interactions, GC B cells are about equally likely to organise BCR:antigen complexes into peripheral clusters and centralised clusters. It would be informative to have, in the same study (Figure 7), a comparison with GC B cells stimulated with antigen alone. The reason is that other studies investigating GC B cell synapse architecture did not quantify antigen organisation in this way, so it is difficult to make comparisons with previous work. It would also be very useful to see how the actomyosin network is organised in GC B cells exhibiting different synaptic architectures (i.e. peripheral versus central clusters), especially given the critical role of myosin IIa activity in GC B cell antigen affinity discrimination. Additionally, while it is a very interesting observation that LFA-1:ICAM-1 interactions may affect GC B cell synapse organisation, it is not clear whether this has an impact on cellular function. For instance, does antigen and actomyosin organisation in GC B cell synapses contribute to differences in signalling or traction force generation? In the introduction the authors suggest that actomyosin arcs contribute to antibody affinity maturation (line 87-88), but without functional studies to support this claim I think it is too speculative.

3) mDia1 miRNA data requires supporting experiments with rescue of the phenotype upon re-expression or, at minimum, scrambled controls. Also the suggestion that mDia1 nucleates the F- actin at the distal edge spikes remains a hypothesis that would be easy to test with immunofluorescence stainings. Can you say or hypothesize how LFA-1 activates mDia1, and if ROCK is important for that, did you try experiments using the Y27632 compound to block it (optional)?

[Editors’ note: further revisions were suggested prior to acceptance, as described below.]

Thank you for submitting your article "A B cell actomyosin arc network couples integrin co-stimulation to mechanical force-dependent immune synapse formation" for consideration by *eLife*. Your article has been reviewed by 3 peer reviewers, and the evaluation has been overseen by a Reviewing Editor and Anna Akhmanova as the Senior Editor. The following individual involved in review of your submission has agreed to reveal their identity: Darius Vasco Köster (Reviewer #3).

Essential revisions:

The reviewers and editors appreciate that manuscript is improved by the revisions, but there remained concerns about the evidence for the role of mDia in forming arcs and the lack of functional data or consideration of function in the discussion.

1. The appropriate controls need to be included to publish the siRNA data on mDia1. As the formin inhibitor is known to be problematic, strengthening the siRNA experiment is necessary to make the conclusion.

2. The definition of an arc still needs to be clarified and relate to the distribution of antigen receptor-antigen complexes. In cells with a more peripheral antigen distribution are their fewer arcs in the interface with the substrate?

3. The parallels between the earlier analysis of myosin arcs in T cells and this study are remarkable, but a discussion of how these compare considering the distinct functions of the two cell types is lacking. A discussion of this comparison could also be helpful to bring more consideration of function to the paper.

*Reviewer #1 (Recommendations for the authors):*

The authors have addressed most of my comments. There is a point raised in my original review that the authors addressed with some modifications/additions to the text, but I think additional analysis of existing data would really strengthen the paper. This is summarised below.

1) The authors suggest that GC B cells with actomyosin arcs are likely those that centralise antigen, but further data analysis is needed to support this claim. Specifically, I think the authors should quantitate the overlap of GC B cells that centralise antigen and those that have actomyosin arcs. Additionally, to give better insight into how actomyosin helps to organise antigen in the synapse, it would be informative to include example images of actomyosin organisation in GC B cells that have peripheral antigen clusters and microclusters (to complement Figure 7, E1-E3) alongside the images already shown of a GC B cell that has centralised antigen (Figure 7, C1-C4). The authors are likely to already have these data in hand so it should not require additional experiments. I think these additions to Figure 7 would strengthen the paper by giving more insight into how antigen organisation in the synapse is dependent on the presence of actomyosin arcs.

*Reviewer #2 (Recommendations for the authors):*

Unfortunately, I feel that the authors have not really acted upon the concerns of the reviewers. No new data, analysis or even any considerable discussion points have been added. The concerns about manual segmentation, yes/no -type of analysis of the actin arcs, and the limited description of the breadth of data remain. Also, siRNA silencing data without scrambled controls or rescue experiments does not belong to today's high quality cell biological publications. This lack of rigor in some experimental parts without efforts to improve it is concerning.

*Reviewer #3 (Recommendations for the authors):*

The authors have considered and included a number of the reviewer comments in their revised version of the article 'A B cell actomyosin arc network couples integrin co-stimulation to mechanical force-dependent immune synapse formation'. The manuscript has gained in clarity and the figures and supplement data underline strongly the findings and claims made by the authors that LFA-1 ligation plays and important role in B-cell activation through the formation of actin arcs that push engaged ICAM-1 clusters towards the cSMAC.

However, one weakness is the lack of additional data to support the findings on the role of mDia1 as the use of scrambled miRNA or immunostaining for mDia1 would be typical control experiments here. The response to reviewers also does not provide any further information why these experiments could not be performed.

Apart from that there are a few points that should be addressed/ clarified:

– line 312 onwards (about kymograph analysis): the image provided does not show convincingly how the velocities could be measured with such precision (2 digit precision with 21 data points). E.g. I cannot make out a clear line for velocity calculation in the dSMAC region of Figure 4-Figure sup 1 A2, and the lines in the pSMAC show a variety of slopes that would result in a higher standard deviation. How many readings from each kymograph were used per cell? Does the bar diagrams in Figure 4-Figure sup1 compare the average values of each cell or compare the measurements from all cells pooled together?

line 951-953 and methods section (line 820-822) (about anisotropy measurements): since the number of measurements differs between the two conditions, it would be better to represent the histogram in relative numbers (frequency or probability) instead of absolute numbers. in addition, it would be good if the description in the methods section could be a bit more insightful, e.g. what criteria were followed to draw ROIs, did you draw ROIs of similar size in the dSMAC and pSMAC regions (the box drawing seems a bit random in the example image, which is puzzling)? It would also be good to indicate that the anisotropy measurement is based on the intensity gradients between pixels.

line 970-971: please add to the legend that the red line indicates the average orientation of the filaments in the ROI.

Figure 5 – sup Figure 2: Within the anti-IgM zones seem to be clusters of phalloidin that are void of anti-IgM signal. is that just an odd effect of the interplay between the fluorescent markers or do these zones have a biological meaning, i.e. are these zones containing some other proteins that bind strongly to actin and exclude the anti-IgM?

---

## [Author Response]

Essential revisions:Your paper is of interest to immunologists studying mechanisms of lymphocyte activation and scientists in the broader field of cell mechanics. The work provides new insight into the cooperation among receptors, the actin cytoskeleton, and myosin motors that is required for the formation of a B cell immune synapse. The data support the key claims of the manuscript.There are a few instances where more complete analysis and presentation of results would strengthen the paper.1) The requirement for LFA-1:ICAM-1 ligation in the formation of the actomyosin arcs is not clear. The authors observe that ~30% of B cells form actomyosin arcs with anti-IgM stimulation only (Figure 1). Does LFA-1:ICAM-1 ligation simply stabilise the arcs and therefore make their appearance more likely, or does it promote the formation of a distinct actomyosin network with unique functions? The images and videos selected to represent cells stimulated with anti-IgM only (Figure 1; Videos 1A and 1B) seem form a highly branched actin network throughout the synapse, but it would be informative to see cells having the actomyosin arcs for comparison. Since B cells stimulated with anti- IgM alone are capable of signalling and centralising antigen, it would be interesting to know whether and how these two populations (with and without arcs) differ.

We thank the reviewers for their questions regarding this central aspect of our study. In response to the reviewers’ statement “The requirement for LFA-1:ICAM-1 ligation in the formation of the actomyosin arcs is not clear”, our results state that “Consistently, scoring B cells for the presence of a discernable actin arc network showed that the addition of ICAM-1 increases the percentage of such cells from ~30% to ~70% (Figure 1G).” Importantly, we then state that “dynamic imaging showed that the arcs in cells engaged with anti-IgM alone are typically sparse and transient (Videos 1A and 1B), while those in cells engaged with both anti-IgM and ICAM-1 are dense and persistent (Videos 2A and 2B).” To emphasize this point, which we think is clear when comparing Videos 1A/1B to Videos 2A/2B, we have now added the following two sentences to the text: “In other words, when B cells receiving only anti-IgM stimulation do form discernable arcs (see, for example, those marked by magenta arrows in Figure 1A and 1B), they are much sparser and less robust than those formed by cells also receiving ICAM-1 stimulation. Moreover, we never saw even one B cell receiving anti-IgM stimulation alone that possessed a robust actin arc network.” Please note that the magenta arrows in Figure 1A and 1B were added upon revision. In summary, the cell shown in Figure 1E, which lacks discernable arcs, is representative of ~70% of anti-IgM stimulated cells, while the cell shown in Figure 1F, which possesses a robust arc network, is representative of ~70% of anti-IgM+ICAM-1 stimulated cells.

We would also like to address what we think is a misunderstanding regarding our images in Figure 1, as reflected in reviewer 1’s statement: “The images and videos selected to represent cells stimulated with anti-IgM only (Figure 1; Videos 1A and 1B) seem form a highly branched actin network throughout the synapse”. The outer, Arp2/3-generated, branched network comprising the dSMAC/lamellipodium in primary B cells is really quite thin under both stimulation conditions (please see Figure 1, E1, E2, F1 and F2). In other words, we would not characterize the region between this thin, outer, canonical branched actin network and the central actin hypodense area (i.e. the region corresponding to the pSMAC) in B cells engaged with anti-IgM alone as “a highly branched actin network throughout”. We described it in the text as “a highly disorganized mixture of short actin filaments/fibers and actin foci”. While it likely contains some branched filaments, it is not a canonical branched actin network like the one comprising the dSMAC. Indeed, it is a lot like the mixture of actin asters, actin foci, branched actin and linear filaments described in Hela cells using the same imaging technique ((Fritzsche et al., 2017); we have now cited this paper). Of note, A20 B cells make a much bigger branched actin/dSMAC/lamellipodium than do primary B cells (compare the image of the representative A20 B cell in Figure 1J to the various images of primary B cells in this figure). Interestingly, this difference between immortalized cells and primary cells is conserved in T cells, as Jurkat T cells make a much bigger branched actin/dSMAC/lamellipodium than do primary T cells (Murugesan et al., JCB 2016).

Although the reviewers did not specifically comment on why only ~70% of primary B cells engaged with both anti-IgM and ICAM-1 make actomyosin arcs, we note that this is also the case for both Jurkat T cells and primary T cells (Murugesan et al., JCB 2016). We do not know why the number does not go to 100%, but the ~70% limit is the case for both B cells and T cells. Of note, in unpublished work we see that LFA-1 ligation also promotes actomyosin arc formation in T cells.

With regard to the reviewers’ question “Does LFA-1:ICAM-1 ligation simply stabilize the arcs and therefore make their appearance more likely, or does it promote the formation of a distinct actomyosin network with unique functions?”, we think that ICAM-1 engagement likely leads to the strong activation of RhoA, which then serves to drive both the formation of actin arcs by recruiting, unfolding, and activating mDia at the plasma membrane, and the stabilization and concentric organization of these arcs by activating myosin 2A filament assembly and contractility. In other words, we think ICAM-1 engagement leads simultaneously to the creation and stabilization/organization of the arcs. While it is true that BCR stimulation alone activates RhoA signaling to some extent (see Saci and Carpenter, Mol Cell 2005 and Caloca *et al.,* J Biol Chem 2008), and that this may account for the sparse actin arcs seen in cells stimulated with anti-IgM alone, it is likely that RhoA signaling is more robust with the addition of integrin co-stimulation (Lawson and Burridge, 2014) and that this would promote the creation of the actomyosin arcs seen in these cells. That said, without independent measures of the creation and stabilization/turnover of the arcs, we cannot gauge the relative significance of creation versus stabilization/turnover in determining the steady state amount of arcs. To address this limitation, we have added the following sentence to the section of the Discussion dealing with integrin-dependent signaling pathways leading to actomyosin arc formation: “Finally, future studies should also seek to clarify the extent to which integrin ligation promotes the formation of actomyosin arcs by driving their creation versus stabilizing them once created.

With regard to the reviewers’ comment that “B cells stimulated with anti-IgM alone are capable of signalling and centralising antigen” we would like to emphasize that our study focuses on B cell immune synapse formation under limiting antigen conditions, where a previous study (Carrasco *et al.,* Immunity 2004) and our data in Figure 4 – supplement 1 show that the impairments in BCR signaling and antigen centralization seen under this condition are rescued by integrin co-stimulation. We expand upon these findings by showing in Figures 5 and 6 that this integrin-dependent rescue of antigen centralization and BCR signaling requires actomyosin. In other words, the actomyosin arc network described here is required for integrin co-stimulation to promote antigen centralization and signaling under limiting antigen conditions. We agree with the reviewer that under non-limiting antigen conditions B cells can signal and centralize antigen in the absence of ICAM-1. That said, these high levels of BCR stimulation are probably not as physiological as limiting BCR stimulation. Finally, our data in Figure 5 – supplement 2 shows that antigen centralization in primary B cells receiving non-limiting anti-IgM stimulation alone is also significantly impaired when myosin is inhibited. This suggests that cells receiving high levels of BCR stimulation employ myosin in some fashion to drive antigen centralization. We now close the section describing these results with the following statement:

“That said, additional experiments should help define exactly how myosin contributes to antigen centralization in B cells receiving only strong anti-IgM stimulation."

Finally, and most generally, we avoided the use of the word “requirement” as in the reviewer’s statement “the requirement for LFA-1:ICAM-1 ligation in the formation of the actomyosin arcs is not clear”. Given that some B cells receiving only anti-IgM stimulation create arcs (albeit sparse and transient), we were careful to say throughout the text that ICAM-1 engagement “promotes” actomyosin arc formation. We think our evidence for this is compelling.

2) The observation that some GC B cells centralise antigen is very interesting, but there are a few aspects of this investigation that should be expanded upon. The authors show that with LFA-1:ICAM-1 interactions, GC B cells are about equally likely to organise BCR:antigen complexes into peripheral clusters and centralised clusters. It would be informative to have, in the same study (Figure 7), a comparison with GC B cells stimulated with antigen alone. The reason is that other studies investigating GC B cell synapse architecture did not quantify antigen organisation in this way, so it is difficult to make comparisons with previous work. It would also be very useful to see how the actomyosin network is organised in GC B cells exhibiting different synaptic architectures (i.e. peripheral versus central clusters), especially given the critical role of myosin IIa activity in GC B cell antigen affinity discrimination. Additionally, while it is a very interesting observation that LFA-1:ICAM-1 interactions may affect GC B cell synapse organisation, it is not clear whether this has an impact on cellular function. For instance, does antigen and actomyosin organisation in GC B cell synapses contribute to differences in signalling or traction force generation? In the introduction the authors suggest that actomyosin arcs contribute to antibody affinity maturation (line 87-88), but without functional studies to support this claim I think it is too speculative.

We thank the reviewer for their comments and suggestions regarding our GC data. Our sole purpose in performing the experiments in Figure 7 was to see if GC B cells can also make actomyosin arcs. We did this because recent papers and reviews state that the organization and dynamics of actin at GC B cell synapses are completely different from the organization and dynamics of actin at naive B cells synapses. As such, these initial observations are meant to add to previous work on GC B cells rather than generate direct comparisons. The reviewers appear to agree that the data in Figure 7 shows convincingly that a subset of GC B cells can make actomyosin arcs that are indistinguishable in appearance from those formed by naive B cells (so the specific claim we are making does not “require additional supporting data”). Rather, the reviewers request that we expand on the data in Figure 7 in several ways, some of which we had already mentioned in the Discussion (“While additional work is required to prove that the subset of GC B cells with actomyosin arcs are the ones that centralize antigen, this seems likely given our evidence here that actomyosin arcs drive antigen centralization in naïve B cells.”, and “Future work will also be required to understand why GC B cells vary with regard to actomyosin organization and the ability to centralize antigen 18 (e.g. dark zone versus light zone GCs)”). In addition to these statements, we now end the section describing the results in Figure 7 with the following statement:

“We note, however, that our conclusions regarding actomyosin arcs in GC B cells require additional supporting data that include testing the ICAM-1 dependence of actomyosin arc formation and quantitating the contributions that this contractile structure makes to GC B cell traction force, signaling, and antigen centralization.”

With regard to the reviewers concerns indicated by their comment “In the introduction the authors suggest that actomyosin arcs contribute to antibody affinity maturation (line 87-88), but without functional studies to support this claim I think it is too speculative”, we have changed the relevant sentence to “Finally, we show that germinal center (GC) B cells can also create this actomyosin structure, suggesting that it may contribute to the functions of GC B cells as well”.

3) mDia1 miRNA data requires supporting experiments with rescue of the phenotype upon re-expression or, at minimum, scrambled controls. Also the suggestion that mDia1 nucleates the F- actin at the distal edge spikes remains a hypothesis that would be easy to test with immunofluorescence stainings. Can you say or hypothesize how LFA-1 activates mDia1, and if ROCK is important for that, did you try experiments using the Y27632 compound to block it (optional)?

We now end the section on mDia1 with the following sentence: “We note, however, that this latter conclusion requires additional supporting data, including quantitative imaging of cells treated with a scrambled miRNA control and immunostaining for mDia1.”

With regard to ROCK, we did do one experiment that showed Y27632 disrupts actin organization much like BB does. We would prefer, if possible, to save this data (and the required repeats) for an ongoing study exploring the signaling pathways leading to actomyosin arc formation following LFA-1 ligation.

References

Babich, A., and Burkhardt, J. K. (2011). Lymphocyte signaling converges on microtubules. Immunity, 34(6), 825-827. Retrieved from https://www.ncbi.nlm.nih.gov/pubmed/21703536. doi:10.1016/j.immuni.2011.06.004

Burnette, D. T., Shao, L., Ott, C., Pasapera, A. M., Fischer, R. S., Baird, M. A.,... Lippincott-Schwartz, J. (2014). A contractile and counterbalancing adhesion system controls the 3D shape of crawling cells. J Cell Biol, 205(1), 83-96. Retrieved from https://www.ncbi.nlm.nih.gov/pubmed/24711500. doi:10.1083/jcb.201311104

Freeman, S. A., Lei, V., Dang-Lawson, M., Mizuno, K., Roskelley, C. D., and Gold, M. R. (2011). Cofilin-mediated F-actin severing is regulated by the Rap GTPase and controls the cytoskeletal dynamics that drive lymphocyte spreading and BCR microcluster formation. Journal of Immunology, 187(11), 5887-5900. Retrieved from https://www.ncbi.nlm.nih.gov/pubmed/22068232. doi:10.4049/jimmunol.1102233

Fritzsche, M., Li, D., Colin-York, H., Chang, V. T., Moeendarbary, E., Felce, J. H.,... Eggeling, C. (2017). Self-organizing actin patterns shape membrane architecture but not cell mechanics. Nat Commun, 8, 14347. Retrieved from https://www.ncbi.nlm.nih.gov/pubmed/28194011. doi:10.1038/ncomms14347

Hammer, J. A., Wang, J. C., Saeed, M., and Pedrosa, A. T. (2019). Origin, Organization, Dynamics, and Function of Actin and Actomyosin Networks at the T Cell Immunological Synapse. Annual Review of Immunology, 37, 201-224. Retrieved from https://www.ncbi.nlm.nih.gov/pubmed/30576253. doi:10.1146/annurev-immunol-042718-041341

Kwak, K., Quizon, N., Sohn, H., Saniee, A., Manzella-Lapeira, J., Holla, P.,... Pierce, S. K. (2018). Intrinsic properties of human germinal center B cells set antigen affinity thresholds. Science Immunology, 3(29). Retrieved from https://www.ncbi.nlm.nih.gov/pubmed/30504208. doi:10.1126/sciimmunol.aau6598

Lawson, C. D., and Burridge, K. (2014). The on-off relationship of Rho and Rac during integrin-mediated adhesion and cell migration. Small GTPases, 5, e27958. Retrieved from https://www.ncbi.nlm.nih.gov/pubmed/24607953. doi:10.4161/sgtp.27958

Murugesan, S., Hong, J., Yi, J., Li, D., Beach, J. R., Shao, L.,... Hammer, J. A. (2016). Formin-generated actomyosin arcs propel T cell receptor microcluster movement at the immune synapse. Journal of Cell Biology, 215(3), 383-399. Retrieved from https://www.ncbi.nlm.nih.gov/pubmed/27799367. doi:10.1083/jcb.201603080

Nowosad, C. R., Spillane, K. M., and Tolar, P. (2016). Germinal center B cells recognize antigen through a specialized immune synapse architecture. Nature Immunology, 17(7), 870-877. Retrieved from https://www.ncbi.nlm.nih.gov/pubmed/27183103. doi:10.1038/ni.3458

Tee, Y. H., Shemesh, T., Thiagarajan, V., Hariadi, R. F., Anderson, K. L., Page, C.,... Bershadsky, A. D. (2015). Cellular chirality arising from the self-organization of the actin cytoskeleton. Nat Cell Biol, 17(4), 445-457. Retrieved from https://www.ncbi.nlm.nih.gov/pubmed/25799062. doi:10.1038/ncb3137

Tojkander, S., Gateva, G., Husain, A., Krishnan, R., and Lappalainen, P. (2015). Generation of contractile actomyosin bundles depends on mechanosensitive actin filament assembly and disassembly. eLife, 4, e06126. Retrieved from https://www.ncbi.nlm.nih.gov/pubmed/26652273. doi:10.7554/eLife.06126

Wang, J. C., and Hammer, J. A. (2020). The role of actin and myosin in antigen extraction by B lymphocytes. Seminars in Cell and Developmental Biology, 102, 90-104. Retrieved from https://www.ncbi.nlm.nih.gov/pubmed/31862219. doi:10.1016/j.semcdb.2019.10.017

Yi, J., Wu, X. S., Crites, T., and Hammer, J. A., 3rd. (2012). Actin retrograde flow and actomyosin II arc contraction drive receptor cluster dynamics at the immunological synapse in Jurkat T cells. Molecular Biology of the Cell, 23(5), 834-852. Retrieved from https://www.ncbi.nlm.nih.gov/pubmed/22219382. doi:10.1091/mbc.E11-08-0731

[Editors' note: further revisions were suggested prior to acceptance, as described below.]

Essential revisions:The reviewers and editors appreciate that manuscript is improved by the revisions, but there remained concerns about the evidence for the role of mDia in forming arcs and the lack of functional data or consideration of function in the discussion.1. The appropriate controls need to be included to publish the siRNA data on mDia1. As the formin inhibitor is known to be problematic, strengthening the siRNA experiment is necessary to make the conclusion.

We thank the reviewers and the editor, and have now included two nontargeting miRNA controls, both of which had no effect on the formation actin arcs in A20 cells (please see new Figure 2 —figure supplement C1-C4 and D1-D4). Of note, this result is consistent with the fact that three different mDia1 miRNAs abrogated arc formation, as eliciting the same phenotype with multiple RNAi sequences is generally accepted as good evidence that the phenotype is not due to off-target effects.

Regarding our use of the pan formin inhibitor SMIFH2, which has been reported to inhibit myosin 2 as well as formins, it is important to note that myosin inhibition and formin inhibition have very different effects on arc actin in both T cells (Murugesan JCB 2016) and B cells (reported here): while arc actin disappears when formin activity is inhibited or depleted, arc actin forms when myosin is inhibited but is no longer organized into concentric structures (see Figure 3G). We see the former not the latter in our SMIFH2 experiment (see Figure 2 D2), consistent with a block in arc actin formation downstream of a block in formin activity.

2. The definition of an arc still needs to be clarified and relate to the distribution of antigen receptor-antigen complexes. In cells with a more peripheral antigen distribution are their fewer arcs in the interface with the substrate?

We thank the reviewers and the editor for the opportunity to further clarify how we defined and scored arcs, and to further highlight the major difference between primary B cells that do or do not receive ICAM-1-co-stimulation. With regard to the yes/no scoring of arcs in Figure 1, Panel G, we revised the text to make it clearer that the scores are for the presence of *any* discernible arcs. So, B cells like the ones shown in Figure 1A and B, which received only anti-IgM stimulation, and that have only a couple of arc-like actin filaments in their entire synapse, were scored as yes, just like the B cells in Figure 1C and D, which received both anti-IgM and ICAM-1 stimulation, and that have very robust actin arc networks. We scored this way to be as conservative as possible (and even then, the difference between anti-IgM and anti-IgM plus ICAM-1 was P<0.01). Importantly, in the revised text we then state that “static and dynamic imaging showed that the arcs in cells engaged with anti-IgM alone are sparse and transient (Figure 1A, B; Videos 1A and 1B), while those in cells engaged with both anti-IgM and ICAM-1 are dense and persistent (Figure 1C, D; Videos 2A and 2B)”. To further stress this important point, we added two sentences to the revised manuscript. The first is: “In other words, when B cells receiving only anti-IgM stimulation do form discernable arcs (see, for example, those marked by magenta arrows in Figure 1A, B), they are much sparser and less robust than those formed by cells also receiving ICAM-1 stimulation.” The second is: “Moreover, we could not find any B cells receiving anti-IgM stimulation alone that possessed a robust actin arc network”. We also added the following sentence to the Figure 1 legend:

“Of note, the cell shown in E1/E2 is representative of ~70% of anti-IgM stimulated cells, while the cell shown in F1/F2 is representative of ~70% of anti-IgM+ICAM-1 stimulated cells.” In terms of providing a grey scale quantitation of arcs by counting them in our TIRF-SIM images, we hope the reviewers would agree that doing this is not feasible. That said, we have now included in new Figure 1 —figure supplement 1A1-A3 measurements of the degree of alignment of actin filaments in the pSMAC of B cells stimulated with anti-IgM alone versus B cells stimulated with both anti-IgM and ICAM-1. These anisotropy measurements provide numeric support for our overall conclusion that ICAM-1 co-stimulation promotes the formation of an organized actin arc network in the pSMAC. The text describing this new data reads: “Consistently, measurements of the degree of alignment between actin filaments in the pSMAC portion of B cells stimulated with anti-IgM alone versus both anti-IgM and ICAM-1, which were made using FibrilTool (29), revealed a dramatic shift towards more organized pSMAC actin when ICAM-1 is included (Figure 1 —figure supplement 1A1-A3; see the figure legend for details).”

With regard to how the actin arcs relate to the distribution of antigen, we showed that (i) actin arcs surround centralized antigen in B cells on planar lipid bilayers containing anti-IgM and ICAM-1 (Figure 4A1-A3B), (ii) that the speeds with which antigen clusters and actin arcs move inward across the pSMAC match (Figure 4C, 4D; Figure 4 —figure supplement 1A1-A3), (iii) that individual actin arcs sweep individual BCR antigen clusters inward (Figure 4D-G), and (iv) that the ability of ICAM-1 ligation, which we established promotes the formation of actomyosin arcs, to promote antigen centralization under conditions of limiting antigen requires myosin contractility (Figure 5). Finally, in new data added to the section on GC B cells (see new Figure 7– supplement 1A1-A3 and new Videos 12 and 13) we show that PLB-engaged GC B cells exhibiting centralized antigen show a pSMAC-like accumulation of GFP-M2A surrounding much of the centralized antigen. GC B cells exhibiting peripheral antigen clusters or microclusters, on the other hand, showed no obvious pattern to the distribution of GFP-M2A at the synapse. Moreover, the synapses with peripheral antigen clusters exhibited less total GFP-M2A than the synapses with centralized antigen (new Figure 7– supplement 1B), and the signals that were present appeared quite transient (new Videos 12 and 13). These new results are described in more detail below in our response to Reviewer #1.

In summary, we think our evidence here that the co-stimulatory effect provided by integrin ligation requires the formation of a contractile actomyosin arc network is important for a number of reasons. First, it provides mechanistic insight into the seminal discovery by Carrasco and colleagues that engaging the B cell’s integrin LFA-1 promotes B cell activation and IS formation when membrane-bound antigen binds the BCR weakly or is presented at low density. Second, our results provide a counterpoint to the results in a 2018 *eLife* paper using A20 B cells that concluded myosin has no obvious role in either B cell activation or synapse formation. We think this difference highlights the importance of studying primary B cells. Third, our results show that an actomyosin arc network, not the Arp2/3-complex-dependent branched network, is the major actin network in primary B cells receiving integrin co-stimulation. Fourth, our evidence that myosin promotes B cell activation and synapse formation under conditions of limiting antigen likely has physiological relevance, as the response of follicular B cells to membrane-bound antigen early in the immune response typically involves antigens that bind the BCR weakly, where integrin co-stimulation from ICAM-1 on surface of the APC and subsequent actomyosin arc formation would promote B cell activation. Our results may also have physiological relevance for GC B cells given that the selection of GC B cells with higher affinity BCRs is thought to depend to a significant extent on their ability to gather antigen in the context of strong competition for limiting antigen presented by follicular dendritic cells. Finally, for those interested in defining the roles of actin and myosin in antigen extraction by the B cells, our results highlight the importance of including integrin ligation in understanding the mechanistic basis for this essential B cell function. We think this “reset” is particularly important as several recent studies addressing the mechanism of antigen extraction, including one in *eLife*, did not include a ligand for LFA-1 in their assays. Of note, all of the above points are included at various places in the revised manuscript.

3. The parallels between the earlier analysis of myosin arcs in T cells and this study are remarkable, but a discussion of how these compare considering the distinct functions of the two cell types is lacking. A discussion of this comparison could also be helpful to bring more consideration of function to the paper.

We thank the reviewers and the editor for asking us to speculate about how T cells and B cells might use the force generated by a similar contractile synaptic structure to perform their distinct functions. To address this important question, we revised the second half of the Discussion by consolidating the points made in the previous version and adding specific ideas regarding how T cells and B cells might use the force generated by the same contractile synaptic structure to perform their distinct functions. Here is the relevant portion of the revised Discussion:

“The actomyosin arcs described here in B cells and the actomyosin arcs described previously in T cells (27) have a great deal in common as regards their formation, organization and dynamics (7, 38). Consistently, this contractile structure supports a number of synaptic processes that are shared by these two cell types, including antigen centralization, proximal signaling, and the formation of an adhesive ring in the medial portion of the IS. A major question, then, is how these two immune cell types harness the force generated by this shared contractile structure to perform their unique functions, i.e. target cell killing by the T cell and antibody creation by the B cell. Stated another way, how does the T cell use the force generated by this contractile structure to support the effectiveness of an exocytic event (lytic granule secretion), while the B cell uses the force to support the effectiveness of an endocytic event (antigen extraction and uptake). With regard to T cells, a seminal study by Basu and colleagues (83) showed that actomyosin-dependent forces placed on the target cell membrane by the T cell increase the efficiency of target cell killing by straining the target cell membrane in such a way as to increase the pore-forming activity of perforin. One clear goal, therefore, is to determine if the T cell’s actomyosin arcs are responsible for creating this strain. Imaging the actomyosin arcs during the process of target cell killing, and blocking the force they generate just prior to lytic granule secretion, should reveal their contribution to this essential effector function.

The idea that B cells would use the actomyosin arcs identified here to support the extraction and endocytic uptake of membrane-bound antigens stems from the seminal work of Tolar and colleagues, who showed that M2A plays an important role in antigen extraction (63, 84, 85). These authors also presented evidence that M2A-dependent pulling forces select for BCRs with higher affinity for antigen, as such interactions survive the myosin-dependent strain placed on them, resulting in antigen extraction ((63, 84); for review see (7) and (86)). That said, a recent, imaging-based effort to define the mechanism by which antigen is extracted did not provide clear insight into how M2A contributes to this process. Specifically, Roper et al., (87) reported that the synapses of naïve B cells bound to antigen-bearing plasma membrane sheets (PMSs) are composed of a dynamic mixture of actin foci generated by the Arp2/3 complex and disorganized linear filaments/fibers generated by a formin. While static images showed little co-localization between the actin foci and antigen clusters, dynamic imaging suggested that the foci promote antigen extraction (although formin activity was also required). Based on these and other observations, Roper et al., concluded that naive B cells use a foci-filament network to drive force-dependent antigen extraction (87). How M2A contributes to this force was unclear, however, as M2A (visualized using an antibody to the phosphorylated form of M2A’s RLC) did not co-localize with either actin structure (87). Moreover, neither actin structure was affected by BB treatment. These two findings are notably at odds with our findings that M2A (visualized by endogenous tagging of the M2A heavy chain) co-localizes dramatically with actin arcs, and that BB treatment profoundly disrupts the organization of the pSMAC actin arc network. Regarding this discrepancy, we note that the images of synaptic actin presented by Roper et al., look similar to our images of naïve B cells stimulated with anti-IgM alone, where the synapse was also composed of a disorganized and dynamic mixture of actin foci and short actin filaments/fibers. The fact that the PMSs used by Roper et al., did not contain integrin ligands may explain, therefore, why they did not see a more organized synapse containing actomyosin arcs. In the same vein, two other recent studies examined antigen extraction using substrates that lacked integrin ligands (PLBs and PMSs in (64), and acrylamide gels in (48)). Given our results, we suggest that future efforts to define the mechanism by which M2A promotes antigen extraction should follow the myosin as the B cell extracts antigen from an APC, where the B cell’s integrins will be engaged, and where the antigen can be presented in a physiologically relevant way (e.g. opsonized and bound to an Fc or complement receptor). Such efforts will hopefully reveal how the B cell harnesses the forces generated by the actomyosin arcs identified here to drive antigen extraction and uptake.”

Reviewer #1 (Recommendations for the authors):The authors have addressed most of my comments. There is a point raised in my original review that the authors addressed with some modifications/additions to the text, but I think additional analysis of existing data would really strengthen the paper. This is summarised below.1) The authors suggest that GC B cells with actomyosin arcs are likely those that centralise antigen, but further data analysis is needed to support this claim. Specifically, I think the authors should quantitate the overlap of GC B cells that centralise antigen and those that have actomyosin arcs. Additionally, to give better insight into how actomyosin helps to organise antigen in the synapse, it would be informative to include example images of actomyosin organisation in GC B cells that have peripheral antigen clusters and microclusters (to complement Figure 7, E1-E3) alongside the images already shown of a GC B cell that has centralised antigen (Figure 7, C1-C4). The authors are likely to already have these data in hand so it should not require additional experiments. I think these additions to Figure 7 would strengthen the paper by giving more insight into how antigen organisation in the synapse is dependent on the presence of actomyosin arcs.

We thank Reviewer #1 for their positive assessment of our paper, and for their specific suggestion as to how we might further clarify the connection between antigen centralization and the actomyosin arcs we see in a subset of GC B cells. Before discussing new data related to their suggestion, we would like to mention that we revised the text regarding Figure 7C1-C4 to better emphasize how those images support the connection between the actomyosin arcs and antigen:

“Importantly, scoring showed that about one third of glass-engaged GFP-M2A knockin GC B cells exhibited robust accumulation of M2A filaments in the pSMAC (Figure 7B). Similarly, about one third of GFP-M2A knockin GC B cells engaged for 10 minutes with PLBs containing fluorophore-labeled anti-IgM/IgG and unlabeled ICAM-1, and then fixed and stained with phalloidin, exhibited robust accumulation of M2A filaments in the pSMAC (Figure 7C1-C4, and D). Importantly, these actomyosin arcs can be seen to surround antigen accumulated at the center of the synapse (see the white arrows in Figure 7C1, C2 and C4). This finding, together with the fact that the myosin moves centripetally during IS formation (Video 11), suggests that actomyosin arcs can contribute to antigen centralization in GC B cells as well as in naïve B cells.”

We thank Reviewer #1 for their excellent suggestion to include images of actomyosin distribution in GC B cells that have peripheral antigen clusters and microclusters alongside images of actomyosin distribution in GC B cells that have centralized antigen. Towards that end, we have now included representative images of GFP-M2A distribution in all three types of synapses shown in Figure 7E1-E4 in new Figure 7 —figure supplement 1. The addition of these images, the conclusions we draw from them, and the remaining unanswered questions are covered in the following section of the revised manuscript:

“Given these results, we asked if our PLB-engaged mouse GC B cells can centralize antigen. In partial agreement with previous findings (63,64), ~45% of synapses exhibited small to medium sized antigen clusters distributed to varying degrees in the synapse periphery (Figure 7E1, E2 and F). In addition, ~20% of synapses exhibited antigen microclusters spread throughout the synaptic interface (Figure 7E3 and F). Importantly, the remaining ~35% of synapses exhibited highly centralized antigen (Figure 7E4 and F). Images of these synapses showed a pSMAC-like accumulation of GFP-M2A surrounding much of the centralized antigen (Figure 7 —figure supplement 1A1; see also Video 12). Conversely, images of synapses containing either peripheral antigen clusters or microclusters showed no obvious pattern to the distribution of GFP-M2A (Figure 7 —figure supplement 1A2, A3; see also Video 13). Moreover, the synapses containing peripheral antigen clusters exhibited less total GFP-M2A signal than the synapses with centralized antigen (Figure 7 —figure supplement 1B), and the signals appeared more transient (compare Video 13 to Video 12). These results, together with the images in Figure 7C1-C4, argue that GC B cells with centralized antigen (about one third of cells) are the ones that make actomyosin arcs (again, about one third of cells). We conclude, therefore, that GC B cells can make actomyosin arcs and that they likely use this structure to centralize antigen, although the degree to which they do this is considerably less than for naïve B cells. We note, however, that our conclusions regarding GC B cells require additional supporting data that include testing the ICAM-1 dependence of actomyosin arc formation and quantitating the contributions that this contractile structure makes to GC B cell traction forces, signaling, and antigen centralization.”

Finally, we will comply with *eLife*’s COVID policy on revisions by performing these latter experiments at a later date and uploading our findings on the preprint server.

Reviewer #2 (Recommendations for the authors):Unfortunately, I feel that the authors have not really acted upon the concerns of the reviewers. No new data, analysis or even any considerable discussion points have been added. The concerns about manual segmentation, yes/no -type of analysis of the actin arcs, and the limited description of the breadth of data remain. Also, siRNA silencing data without scrambled controls or rescue experiments does not belong to today's high quality cell biological publications. This lack of rigor in some experimental parts without efforts to improve it is concerning.

We thank Reviewer #2 for their comments. With regard to their concerns about the yes/no -scoring of actin arcs, we ask that they please read our response to Essential Revision #2 above. With regard to their concerns about manual segmentation, we ask that they please read our response to Reviewer #3 below. With regard to their comment about the limited description of the breadth of data, we ask that they please read our response to Essential Revision #3 above. Finally, with regard to Reviewer #2’s comments regarding the lack of a scrambled miRNA control, it was not our intention to be dismissive of this reviewer’s very reasonable request, and we apologize for that. As discussed above, our decision to respond to this request by qualifying our conclusions in the revised text and stating that additional experiments are required to solidify the conclusions rather than by performing the experiments was based on *eLife*’s current policy on revisions made in response to COVID-19, which states that “when editors judge that a submitted work as a whole belongs in *eLife* but that some conclusions require a modest amount of additional new data, as they do with your paper, we are asking that the manuscript be revised to either limit claims to those supported by data in hand, or to explicitly state that the relevant conclusions require additional supporting data.” We also think that getting the same phenotype with three different miRNAs, as we did here for mDia1 (loss of arcs), is generally accepted as good evidence that the phenotype is not due to off-site targets. Given this, given that mDia1 is the most abundant formin in B cells, given that it is responsible for forming arcs in T cells (Murugesan et al., 2016), and given *eLife*’s COVID-era revision policy, we thought that our qualification regarding the role of mDia1 in creating the arcs, as encapsulated by the following sentences in our initial revision “Together, these results argue that the pSMAC actin arcs are indeed created by a formin, and that the formin mDia1 plays a major role. We note, however, that this latter conclusion requires additional supporting data, including quantitative imaging of cells treated with a scrambled miRNA control and immunostaining for mDia1”, was reasonable. That said, we have now included two scrambled/nontargeting miRNA controls and modified the text accordingly (please see our response to Essential Revision Request #1).

Reviewer #3 (Recommendations for the authors):The authors have considered and included a number of the reviewer comments in their revised version of the article 'A B cell actomyosin arc network couples integrin co-stimulation to mechanical force-dependent immune synapse formation'. The manuscript has gained in clarity and the figures and supplement data underline strongly the findings and claims made by the authors that LFA-1 ligation plays and important role in B-cell activation through the formation of actin arcs that push engaged ICAM-1 clusters towards the cSMAC.However, one weakness is the lack of additional data to support the findings on the role of mDia1 as the use of scrambled miRNA or immunostaining for mDia1 would be typical control experiments here. The response to reviewers also does not provide any further information why these experiments could not be performed.

We thank Reviewer #3 for their positive assessment of our paper. With regard to their last point (why we did not include a scrambled miRNA control on our first revision), we ask that they please read the first paragraph of our rebuttal and the confidential information below. Importantly, have now included two nontargeting miRNA controls, both of which had no effect on the formation actin arcs in A20 cells (please see new Figure 2 —figure supplement C1-C4 and D1-D4).

Apart from that there are a few points that should be addressed/ clarified:– line 312 onwards (about kymograph analysis): the image provided does not show convincingly how the velocities could be measured with such precision (2 digit precision with 21 data points). E.g. I cannot make out a clear line for velocity calculation in the dSMAC region of Figure 4-Figure sup 1 A2, and the lines in the pSMAC show a variety of slopes that would result in a higher standard deviation. How many readings from each kymograph were used per cell? Does the bar diagrams in Figure 4-Figure sup1 compare the average values of each cell or compare the measurements from all cells pooled together?

First, we agree that the lines within the dSMAC in the kymograph for the primary B cell in Panel A2 are hard to see (especially as compared to A20 cells), so we added some white arrowheads to Panel 2A to point out several of them. We note that this difference between A20 B cells and primary B cells, where A20 cells have a much more robust dSMAC than primary B cells, is also seen for T cells, where Jurkats have a much more robust dSMAC than primary T cells (Murugesan et al., JCB 2016).

With regard to the flow rates in Figure 4 —figure supplement 1A3 and B3, we thank the reviewer for bringing this up as we did not make it clear in the text, legend or Methods what the numbers represent. We took ~7 measurements per cell for each SMAC. The values reported in A3 and B3 are an average of the mean rates per cell (with 21 and 14 cells scored for A3 and B3, respectively). This is now stated in the revised legend to Figure 4 —figure supplement 1, where we make clear that the bar graphs in A3 and B3 show standard error of the means (i.e. not standard deviations).

line 951-953 and methods section (line 820-822) (about anisotropy measurements): since the number of measurements differs between the two conditions, it would be better to represent the histogram in relative numbers (frequency or probability) instead of absolute numbers.

We thank the reviewer for this suggestion and now present the pre-existing anisotropy data comparing control and BB-treated cells, as well as the new anisotropy data comparing cells stimulated with anti-IgM alone versus anti-IgM plus ICAM1, as frequency measurements.

in addition, it would be good if the description in the methods section could be a bit more insightful, e.g. what criteria were followed to draw ROIs, did you draw ROIs of similar size in the dSMAC and pSMAC regions (the box drawing seems a bit random in the example image, which is puzzling)?

We thank the reviewer for seeking clarification regarding how we segmented our TIRF-SIM images. Drawing ROIs encompassing the pSMAC to measure the anisotropy of actin filaments within it, as well as many other measurements in the paper (e.g. the content of F-actin in SMACs), required that we segment synapses into their three SMACs. To accomplish this, we relied on the distinctive appearance of actin in each SMAC. This was easiest for B cells stimulated with both anti-IgM and ICAM-1, where the distinctions between the three SMACs were usually very clear in TIRF-SIM images (see for example Figure 1F1 and F2). Specifically, the thin outer dSMAC was comprised of moderately bright pixels with not much fluctuation in intensity, the medial pSMAC was comprised of bright actin arcs with intervening dim signals, and the central cSMAC was comprised mostly of dim signals. These distinctions were, however, less obvious in cells engaged with anti-IgM alone (see for example Figure 1E1 and E2), although the thin outer dSMAC was still identifiable, and the cSMAC was evident in lower mag images as a central circle with less signal than the area between it and the dSMAC (see for example Figure 1B). The latter area corresponded, then, to the pSMAC in cells stimulated with anti-IgM alone. As requested, we improved our description for how we performed manual segmentation of our TIRF-SIM images by including a version of the explanation above in the revised Methods section.

It would also be good to indicate that the anisotropy measurement is based on the intensity gradients between pixels.

We thank the reviewer for this comment and we have now included the following sentence in the methods “The FibrilTool plugin for ImageJ was used to measure actin arc morphology based on the intensity gradients between pixels as described previously (27, 29).”

line 970-971: please add to the legend that the red line indicates the average orientation of the filaments in the ROI.

We have added this to the figure legend for Figure 1—figure supplement 1. Thank you.

Figure 5 – sup Figure 2: Within the anti-IgM zones seem to be clusters of phalloidin that are void of anti-IgM signal. is that just an odd effect of the interplay between the fluorescent markers or do these zones have a biological meaning, i.e. are these zones containing some other proteins that bind strongly to actin and exclude the anti-IgM?

We thank Reviewer #3 for this question. That said, we cannot at this time weigh in on the biological meaning of the phalloidin clusters they are referring to.